# DenseMLLM: Standard Multimodal LLMs for Dense Prediction

Yi Li [1 2]  Hongze Shen [2]  Lexiang Tang [2]  Xin Li [2 *]  Xinpeng Ding [1]
Yinsong Liu [2]  Deqiang Jiang [2]  Xing Sun [2]  Xiaomeng Li [1 *]

## Abstract

Multimodal Large Language Models (MLLMs) have demonstrated exceptional capabilities in high-level visual understanding. However, extending these models to fine-grained dense prediction tasks, such as semantic segmentation and depth estimation, typically necessitates the incorporation of complex, task-specific decoders and other customizations. This architectural fragmentation increases model complexity and deviates from the generalist design of MLLMs, ultimately limiting their practicality. In this work, we challenge this paradigm by accommodating standard MLLMs to perform dense predictions without requiring additional task-specific decoders. The proposed model is called DenseMLLM, grounded in *the standard architecture with a novel vision token supervision strategy for multiple labels and tasks.* Despite its minimalist design, our model achieves highly competitive performance across a wide range of dense prediction and vision-language benchmarks, demonstrating that a standard, general-purpose MLLM can effectively support dense perception without architectural specialization. This project is available at github.com/Eli-YiLi/DenseMLLM.

## 1. Introduction

Multimodal Large Language Models (MLLMs) have achieved substantial success in vision-language understanding, utilizing a next-token prediction paradigm to unify tasks like VQA, OCR, and grounding via open-ended conversations. However, a critical limitation persists: a general-purpose MLLM framework struggles to accommodate fine-grained dense prediction tasks like semantic segmentation

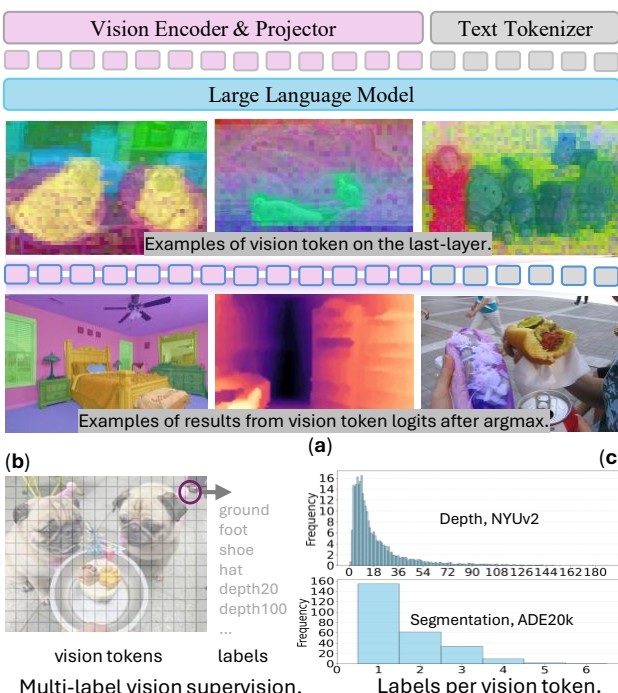

*Figure 1.* **Motivation.** **(a)**: PCA visualization of the hidden states reveals that the vision tokens of the proposed DenseMLLM intrinsically encode fine-grained details. Thus, DenseMLLM achieves high-quality dense predictions (segmentation and depth) directly from vision tokens without task-specific decoders. **(b)**: A single vision token typically represents multiple vocabulary IDs (labels), especially in multi-task scenarios, which contrasts with text tokens with only a single label. **(c)**: These histograms indicate that vision tokens frequently have multiple labels across tasks. It motivates us to propose a new supervision strategy to align vision-text representations effectively for multi-label and multi-task.

and depth estimation. To address this, current approaches often rely on external task-specific decoders (Rasheed et al., 2024; Liu et al., 2026) or task embedding tokens (Tang et al., 2026)). However, these designs fragment the architecture and complicate the framework. Alternative methods that attempt to use a unified vocabulary also face significant drawbacks: DepthLM (Cai et al., 2025) requires many inferences to predict the depth of multiple marked points; polygon-based approaches (Lu et al., 2024; Pramanick et al., 2024) often perform poorly owing to limited coordinate numbers. Consequently, achieving high-quality dense pre-

[1]Department of Electronic and Computer Engineering, The Hong Kong University of Science and Technology, Hong Kong, China [2]Tencent, Youtu-Lab, China. Correspondence to: Xin Li <fujikoli@tencent.com>, Xiaomeng Li <eexmli@ust.hk>.

*Proceedings of the $43^{rd}$ International Conference on Machine Learning*, Seoul, South Korea. PMLR 306, 2026. Copyright 2026 by the author(s).

diction without sacrificing the architectural simplicity of MLLMs remains a largely unsolved challenge.

In our view, with appropriate supervision, standard MLLM architectures utilizing intrinsic vision tokens can be an architecture-free dense predictor without extrinsic decoders, as shown in Fig. 1a. In this paper, we introduce **DenseM-LLM**, a general-purpose MLLM that accommodates dense prediction tasks without relying on task-specific decoders. Our method includes two core designs. (**1**) **Standard MLLM for dense prediction**: We extract dense predictions directly from the vision tokens generated by the MLLM, thereby eliminating the need for specialized external decoders and avoiding the computational overhead of additional inference steps. Specifically, we predict the text of the target categories in a unified vocabulary and use their corresponding token IDs to extract specific logits from the vision tokens. We then apply the argmax to obtain the final predictions. (**2**) **Vision NTP for multi-label**: We transfer the Next-Token Prediction (NTP) paradigm from text to the vision tokens for high-quality dense prediction. Unlike text tokens that possess a single label, vision tokens are inherently multi-semantic and often encompass multiple tasks (see Fig. 1b). To address this fundamental difference, we propose a multi-label vision supervision paradigm and a relevant negative sampling strategy for robust optimization. Building on these principles, we train DenseMLLM, a 4B-parameter model based on a standard ViT, projector, and LLM architecture. It successfully covers diverse vision-language tasks while achieving high-quality dense predictions without any architectural additions.

Attributed to our effective design and large-scale pre-training, DenseMLLM achieves highly competitive performance across both dense prediction and general vision-language tasks, despite using a standard MLLM architecture with no task-specific components. Specifically, it attains 54.2 mIoU on ADE20K for semantic segmentation, outperforming VisionLLM-v2 (Wu et al., 2024) (52.3) without any task-specific head; achieves 87.6 $\delta_1$ on DDAD for depth estimation, beyond DepthLM (Cai et al., 2025) (74.7) without multiple inference passes; and obtains 80.7 cIoU on RefCOCO-val for referring expression segmentation, exceeding UFO (80.0) without the retrieval process. Meanwhile, it matches Qwen3-VL-4B (Bai et al., 2025a) on standard general VL benchmarks, demonstrating that dense visual understanding can be effectively integrated into a general-purpose MLLM. These results demonstrate that dense visual understanding can be freely integrated into a standard general-purpose MLLM with high performance. Our main contributions include:

- Standard MLLM for Dense Prediction: We introduce DenseMLLM, which uses a standard architecture and unified vocabulary for dense prediction tasks without

task-specific decoders.

- Vision NTP for Multi-label: we extend the next-token prediction objective to vision tokens, allowing multiple labels and tasks within a vision token for supervision.

- DenseMLLM: We present a standard Multimodal LLM that excels at both dense prediction tasks and general VL tasks, broadening the capacity scope of the general-purpose foundation model.

## 2. Related Works

**Dense Prediction Tasks.** Dense prediction encompasses tasks requiring fine-grained pixel-level understanding. In this paper, we focus on three tasks: (**1**) Semantic segmentation assigns semantics to every pixel, differing from previous architectures evolved from CNNs (Zhao et al., 2017) to Transformer (Cheng et al., 2022; Xie et al., 2021) and Diffusion architectures (Zhao et al., 2023). (**2**) Depth estimation predicts pixel-wise distance, critical for 3D geometry, including specialist models (Piccinelli et al., 2025; Yang et al., 2024b) and the VLM-base method (Cai et al., 2025). (**3**) Referring expression segmentation (RES, or referring segmentation) bridges vision and language by segmenting objects based on natural language queries (Rasheed et al., 2024; Liu et al., 2026). The above tasks are sufficiently representative to demonstrate that our method is widely applicable to dense prediction tasks.

**Dense Prediction Methods.** The related methods mainly consist of four types. (**1**) Specialist models (conventional task-specific fine-tuned methods) such as Segformer (Xie et al., 2021) and Mask2Former (Cheng et al., 2022), utilize task-specific decoders (e.g., MLP or Pixel Decoders) on segmentation, while models like UniDepth-v2 (Piccinelli et al., 2025) focus on depth estimation without other capabilities. (**2**) Vision Generalist models (e.g., X-Decoder (Zou et al., 2023), SEEM (Mizrahi et al., 2023)) employ shared decoders or contrastive adaptation based on CLIP (e.g., MaskCLIP (Zhou et al., 2022), SAN (Xu et al., 2023)). However, these non-generative frameworks lack the open-ended conversational and reasoning capabilities inherent to modern Large Language Models (LLMs). (**3**) Multimodal LLMs with additions integrate dense prediction into MLLMs generally following two paths. Add-on approaches equip LLMs with external modules: GLaMM (Rasheed et al., 2024) and UniPixel (Liu et al., 2026) append SAM-based decoders, while UFO (Tang et al., 2026) uses special mask embedding tokens requiring extra retrieval steps. GiT (Wang et al., 2024) does not use extra heads, but its parallel decoding demands extra codebases like mmdet. These designs fragment the architecture and complicate the pipeline. (**4**) Standard MLLMs attempt to use the native autoregressive framework but face significant trade-offs: polygon-based methods like

VisionLLM (Wang et al., 2023b) often lack precision, and text-based approaches like DepthLM (Cai et al., 2025) incur high computational costs due to iterative inference in depth estimation. Unlike these, *our method achieves high-quality dense prediction using a purely standard architecture without external decoders or complex decoding schemes.*

**Vision Token Supervision.** Standard MLLMs predominantly apply Next-Token Prediction (NTP) to textual tokens, leaving visual features aligned only via global objectives. Recent attempts to extend NTP to vision tokens typically target high-level tasks like VQA and lack the granularity for dense prediction, such as VT-LLM (Peng et al., 2024), SEA (Yin et al., 2025), etc. (Yoon et al.; Tang et al., 2025; Bao et al.). These methods are still based on the conventional next token prediction objective, where each vision token corresponds to a single vocabulary index. Their difference mainly lies in how the labels for vision tokens are obtained. Although some methods (Zhou et al., 2022; Li et al., 2025b;a) try to extract dense predictions from vision tokens, they lack vision-language alignment as training-free techniques with limited performance. Our approach introduces a multi-label autoregressive loss for vision tokens, effectively handling the ambiguity of token-level semantics and ensuring tight image-text alignment. Different from the related works, *Our method supports multiple labels and tasks, enabling high-quality dense predictions instead of merely providing indirect improvements to the VQA task.*

## 3. Methodology

**Preliminary.** Standard vision-language models generate text responses $\mathbf{Y}$ autoregressively given a vision input $\mathbf{X}_v$ and instruction $\mathbf{X}_{\text{instruct}}$. The training objective follows the Next-Token Prediction (NTP) paradigm, minimizing the negative log-likelihood over a unified vocabulary:

$$\mathcal{L}_{\text{NTP}} = -\sum_{t=1}^{T} \log p(y_t | \mathbf{X}_v, \mathbf{X}_{\text{instruct}}, y_{<t}) . \quad (1)$$

While this paradigm excels at text generation, conventional approaches typically resort to external, task-specific decoders for dense predictions (e.g., segmentation), which fragments the unified architecture. Our goal is to eliminate such add-ons and empower a standard MLLM to inherently function as a dense predictor. Specifically, in Sec. 3.1, we detail how to extract dense predictions directly from the standard MLLM architecture, and in Sec. 3.2, we introduce the supervision strategy for vision tokens with multi-label to achieve high-quality prediction results.

### 3.1. Standard MLLM for Dense Prediction

**Standard Architecture.** As illustrated in Fig. 2, DenseM-LLM adopts a standard MLLM architecture consisting of three core components: (1) a vision encoder based on SigLIP-2 (siglip2-so400m-patch16-naflex) (Tschannen et al., 2025), which incorporates 2D RoPE (Su et al., 2024) for spatial encoding and a hybrid window-global attention mechanism to natively handle variable-length sequences and high-resolution inputs; (2) a vision-language projector that utilizes a $2 \times 2$ spatial merge operation (Bai et al., 2023) to compress visual tokens, followed by a two-layer MLP to map these features into the linguistic embedding space; and (3) The LLM is a 4B parameter transformer evolved from the architecture of (Lu et al., 2026). The above three structures are the classic architectures of visual-to-text MLLMs, ensuring the indirectness of the framework and avoiding additional involvement. This represents that the work in this article is a general and standard structure.

**Dense Prediction.** Our framework generates dense predictions through two core inference mechanisms: (1) determining the active category vocabulary, and (2) deriving results by applying an argmax operation to the vision token logits corresponding to those categories. For *semantic segmentation*, we first predict categories in the image and use their vocabulary IDs to indicate the next argmax, instead of the full large vocabulary set. For *depth estimation*, we discretize the depth range into bins (using linear or logarithmic quantization) and apply argmax across the bin vocabulary.

To handle categories represented by multiple tokens (e.g., sub-word units), we aggregate their scores by averaging the raw vision token logits $\mathbf{Z}$ associated with the set of vocabulary indices $\mathcal{S}_k$ for category $k$. These aggregated logits are reshaped ($\mathcal{R}$) into a spatial grid and upsampled via bilinear interpolation ($\mathcal{I}$) to match pixel resolution. The final dense prediction map $\mathbf{M}$ is derived by taking the argmax ($\mathcal{A}$) over the predicted categories:

$$\mathbf{M} = \mathcal{A}\Big(\mathcal{I}\big(\mathcal{R}\big(\hat{\mathbf{Z}}\big)\big)\Big), \quad \text{where} \quad \hat{\mathbf{Z}}_k = \frac{1}{|\mathcal{S}_k|} \sum_{v \in \mathcal{S}_k} \mathbf{Z}_v . \quad (2)$$

To further improve quality, image zooming strategies can be employed to leverage the model's native resolution for capturing finer details, and optional post-processing is applicable following the common practice (Krähenbühl & Koltun, 2011).

### 3.2. Vision NTP for Multi-label

**Vision Token Supervision.** A core contribution of our work is the autoregressive supervision of vision tokens. We extend the next-token prediction (NTP) paradigm to vision tokens to facilitate a tighter alignment between image and text representations, ensuring robust dense prediction performance rather than restricting supervision solely to textual tokens. Acknowledging that a single vision token often encapsulates multiple semantic labels and task targets in the vocabulary set (see Fig. 2), we employ a multi-label supervision objec-

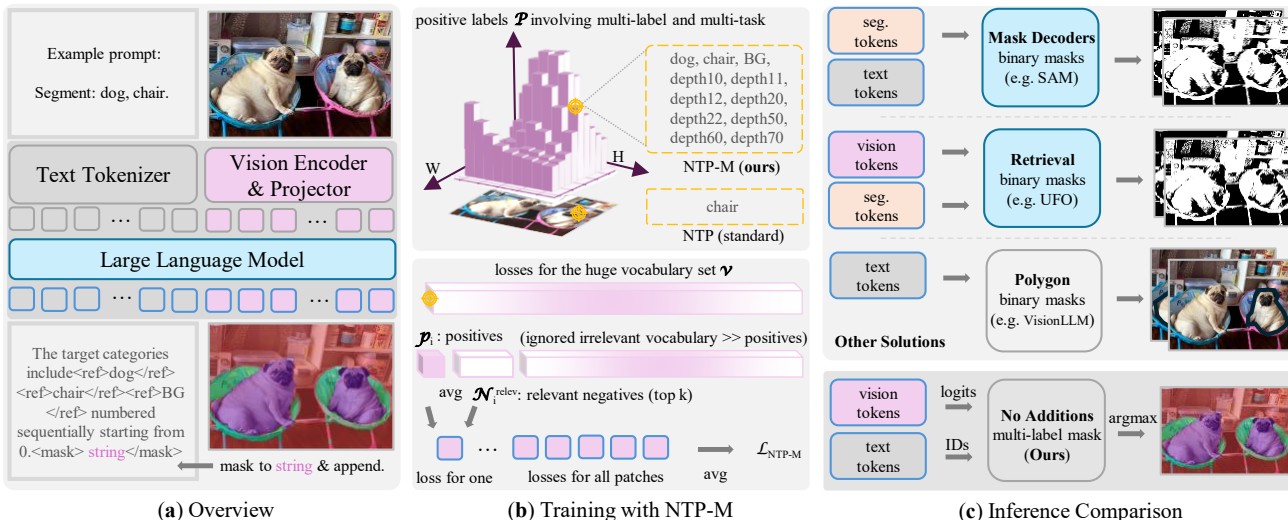

**(a)** Overview         **(b)** Training with NTP-M         **(c)** Inference Comparison

*Figure 2.* **The framework of DenseMLLM**. **(a) Overview**: Employing a standard MLLM architecture (Vision Encoder, Projector, LLM), our model outputs both text responses and dense predictions without specialized heads. **(b) Training with NTP-M**: To handle vision tokens containing multiple semantics (e.g., objects and depth), we propose NTP-M. This multi-label strategy supervises vision tokens against a vocabulary with the proposed relevant negative sampling strategy, extending beyond standard single-label NTP. **(c) Inference Comparison**: Unlike methods relying on external decoders (e.g., SAM-based (Rasheed et al., 2024)), retrieval tokens (e.g., UFO (Tang et al., 2026)), or polygons (Wang et al., 2023b), DenseMLLM requires no additions. We achieve dense prediction by indexing vision logits with text token IDs via argmax, ensuring architectural simplicity.

tive for vision tokens. We term this NTP-M, a variant of the standard NTP adapted for multi-label and multi-task scenarios. Specifically, we transition from a single-label to a multi-label framework by constructing a multi-hot target vector for each vision token, where the token IDs corresponding to all relevant objects, tasks, and potential granularities are set to 1, while the remaining vocabulary indices are assigned 0. Unlike the standard NTP, which models a categorical distribution via Softmax, we model the probability of each token in the vocabulary independently. Consequently, we redefine the generative objective $p(\mathbf{Y}|\mathbf{X}_v, \mathbf{X}_{\text{instruct}})$ as the joint probability of independent Bernoulli trials over the vocabulary:

$$p(\mathbf{Y}|\mathbf{X}_v, \mathbf{X}_{\text{instruct}}) = \prod_{i=1}^{L} \prod_{v=1}^{|\mathcal{V}|} \sigma(\mathbf{Z}_{i,v})^{y_{i,v}} \cdot (1 - \sigma(\mathbf{Z}_{i,v}))^{(1-y_{i,v})} , \quad (3)$$

where $\mathbf{X}_v, \mathbf{X}_{\text{instruct}}$ are the vision token input and instruction (prompt) input, respectively. $L$ denotes the sequence length of vision tokens, $|\mathcal{V}|$ is the vocabulary size, $\sigma(\mathbf{Z}_{i,v})$ represents the sigmoid-activated probability for token $v$ at vision token $i$, and $y_{i,v} \in \{0, 1\}$ indicates the ground truth presence of the corresponding target. Note that $\mathbf{X}_{\text{instruct}}$ can influence the dense prediction process; for example, the prompt before the image guides depth estimation for the depth range of a certain camera.

**Relevant Negative Sampling.** The extensive vocabulary size inherent to MLLMs introduces significant class imbalance, which hinders effective supervision. To address this, we implement a robust multi-label next-token prediction

loss, $\mathcal{L}_{\text{NTP-M}}$, which replaces naive averaging with a strategy based on independent positive-negative averaging and relevant negative sampling.

Specifically, we decouple the loss computation for positive and negative samples to ensure stable convergence. For positive samples, we compute the mean loss across all valid instances. For negative samples, averaging the overwhelming number of irrelevant (low-probability) negatives causes gradient dilution. To prevent this, we rank negative samples by their predicted probabilities $p$ (see Eq. 3) and compute the average loss only for the top-$k$ "relevant" negatives.

This approach differs from standard online hard example mining strategy (Shrivastava et al., 2016), which typically ranks positive and negative samples jointly across the spatial dimension. By operating along the vocabulary dimension and separating the groups, our method prevents positive samples from being overshadowed by the sheer volume of relevant negatives, thereby enhancing convergence efficiency. Additionally, to accommodate incomplete annotations (e.g., missing depth), we employ a validity mask to exclude invalid indices.

Formally, let $p_{i,v} = \sigma(\mathbf{Z}_{i,v})$ denote the Sigmoid-activated probability for vocabulary ID $v$ at vision token $i$, and let $\mathcal{M}_i$ be the set of valid indices given the current task annotations. We define the set of positive indices as $\mathcal{P}_i = \{v \in \mathcal{M}_i \mid y_{i,v} = 1\}$. For negative samples, we define the candidate set as $\mathcal{C}_i = \{v \in \mathcal{M}_i \mid y_{i,v} = 0\}$. From this, the relevant negative set $\mathcal{N}_i^{\text{relev}}$ is derived as the subset of $\mathcal{C}_i$ with size $k$

that maximizes the sum of predicted probabilities (i.e., the top-$k$ relevant negatives):

$$\mathcal{N}_i^{\text{relev}} = \underset{\mathcal{S} \subset \mathcal{C}_i, |\mathcal{S}|=k}{\arg\max} \sum_{v \in \mathcal{S}} p_{i,v} \ . \tag{4}$$

The final loss $\mathcal{L}_{\text{NTP-M}}$ is computed by summing the independent averages over these two sets, thereby balancing the supervision signals from positive targets and the relevant negatives for robust optimization:

$$\mathcal{L}_{\text{NTP-M}} = \sum_{i=1}^{L} \left[ -\frac{1}{|\mathcal{P}_i|} \sum_{v \in \mathcal{P}_i} \log(p_{i,v}) - \frac{1}{k} \sum_{v \in \mathcal{N}_i^{\text{relev}}} \log(1 - p_{i,v}) \right] . \tag{5}$$

## 4. Experiments

### 4.1. Training

**Training Recipe.** The training of DenseMLLM follows a progressive, multi-stage training recipe. This pipeline is structured into four sequential phases: a unified pre-training stage to establish a multimodal foundation (Stage I), a specific annealing stage focused on dense prediction adaptation (Stage II), followed by high-quality supervised fine-tuning (Stage III) and reinforcement learning (Stage IV). Detailed training recipe is provided in Sec. B.

Stage I: Multimodal Foundation Pre-training. This phase establishes the model's fundamental capabilities by merging the language backbone training with multimodal alignment. The pre-trained data involves large scale corpus about commonsense, STEM, and coding data. To endow the model with robust visual perception, we conduct end-to-end training on a comprehensive dataset, mixing high-quality private text with large-scale image-caption pairs and visual-centric task data. This stage balances linguistic proficiency with cross-modal contextual understanding. At the end of this stage, we performed additional pre-training supervision on the vision token, using a codebook learned from the image tokens. Subsequent stages do not involve this.

Stage II: Annealing Pre-training for Versatile Task. This phase serves as the core contribution of our training recipe, focusing specifically on adapting the model for dense prediction tasks. Unlike general multimodal adaptation, this "annealing" stage utilizes a high-quality dataset curated for fine-grained pixel-level understanding, covering tasks such as segmentation, grounding, etc. To maintain the model's general versatility during this stage, we also incorporate auxiliary data such as General VQA and OCR.

Stage III: High-Quality Supervised Fine-Tuning (SFT). This stage aims to refine the model's capability to understand complex instructions, enhance reasoning, and align with human preferences. During this phase, we extend the context window to 32K tokens from 16K. For our core task, we add some open data, flexible prompts, and translated data for a better user experience.

Stage IV: Reinforcement Learning (RL). We follow the setup adopted in DAPO (Yu et al., 2026) to further optimize the model. Crucially, we introduce specific rewards for dense prediction tasks: class-label IoU (label set overlap) for semantic segmentation. Following previous work (Qi et al., 2025), we remove the KL penalty term and adopt FP16 training to ensure faster convergence and stable dynamics.

**Dataset Description.** For the core dense prediction tasks in this paper, the data mainly consists of three types. (1) Semantic segmentation data are collected from massive open-source datasets with masks, and then mapped to text tokens for each vision token, with invalid regions explicitly ignored. (2) Referring expression segmentation data is cropped with padding from many mask instances from open-source datasets, with a random colored box drawn on the image to indicate the target foreground and background (assigned <FG> and <BG> in the vocabulary). (3) Depth estimation applies a linear quantization pipeline (1–1000 bins) for open data with a fixed camera, ignoring label 0 (<custom_0> in the tokenizer vocabulary). We applied augmentations, including random color jitter, crop, aspect ratio, horizontal flip, and resize (0.5-4). The random flip and resize are not applied to the depth data, while adding a random cutout augmentation.

We also prepare a data pipeline for open scenarios. (1) Segmentation: Raw data uses segmentators, while labeled data undergoes class binding. We employ a Copy-Paste strategy to densely place transparent objects. (2) Depth estimation: Raw data utilizes pseudo-labels from open-source models. Labeled data is normalized to a fixed focal length (2000 pixels) using log-uniform quantization (0.5m–100m); inputs with other focal lengths output the relative depth.

Besides the above data, the majority of our training data is constructed based on open-source datasets, synthetic data, and internal resources. Other data involves visual grounding, image caption and knowledge data, optical character recognition (OCR) data, STEM (science, technology, engineering, and mathematics) data, graphical user interface (GUI) data, and pure text data.ing our proprietary internal data sources cannot be disclosed.

**Evaluation.** We evaluate our model across three primary tasks. For semantic segmentation, benchmarks include ADE20k (Zhou et al., 2017), COCO-Stuff (Caesar et al., 2018), Pascal Context (59 foreground classes) (Mottaghi et al., 2014), Cityscapes (Cordts et al., 2016), and Pascal VOC (20 foreground classes), with mean Intersection over Union (mIoU) as the metric. We employ the prompt "Segment: $C$.", where $C$ represents randomly shuffled class names. To demonstrate robustness in real-world scenar-

*Table 1.* **Quantitative comparison on dense prediction tasks.** We evaluate the performance on Semantic Segmentation (mIoU), Depth Estimation ($\delta_1$), and Referring Segmentation (cIoU for RefCOCO, RefCOCO+, RefCOCOg). The table compares our proposed DenseMLLM against Vision Specialist Models, CLIP-based methods, Vision Generalist Models, and other Multimodal LLMs (both with architectural additions and standard setups). "Additions" refers to the usage of additional task-specific decoders, heads, or processes. "×" denotes that the method is not applicable and "-" indicates results are not available. Results in gray indicate task-specific fine-tuning on a single dataset, which usually brings higher results. Best results in each setting are marked in bold.

| Methods | Additions | Semantic Segmentation | | | | | Depth Estimation | | | Referring Segmentation (RefCOCO) | | | | | | | |
| --- | --- | --- | --- | --- | --- | --- | --- | --- | --- | --- | --- | --- | --- | --- | --- | --- | --- |
| | | ADE | COCOstuff | Context59 | Cityscapes | VOC20 | NYUv2 | DDAD | Cityscapes | val | testA | testB | val+ | testA+ | testB+ | val-g | test-g |
| ***Vision Specialist Models*** | | | | | | | | | | | | | | | | | |
| Segformer (MiT-B5) (Xie et al., 2021) | MLP Decoder | 51.0 | 46.7 | - | 82.4 | - | × | × | × | × | × | × | × | × | × | × | × |
| Mask2Former (Swin-L) (Cheng et al., 2022) | Pixel Decoder | 56.0 | - | - | 83.3 | - | × | × | × | × | × | × | × | × | × | × | × |
| UniDepth-v2 (ViT-L) (Piccinelli et al., 2025) | Depth Decoder | - | - | - | - | - | 98.8 | 88.2 | - | × | × | × | × | × | × | × | × |
| SwinMTL (Swin-B) (Taghavi et al., 2024) | MLP Decoder | - | - | - | - | 76.41 | - | - | 92.1 | × | × | × | × | × | × | × | × |
| RVG (ViT-B) (Ouyang et al., 2025) | MLP Decoder | × | × | × | × | × | × | × | × | 79.4 | 81.2 | 77.8 | 69.5 | 75.7 | 63.0 | 71.3 | 72.1 |
| VPD (UNet) (Zhao et al., 2023) | Denoising Decoder | 53.7 | - | - | - | - | 96.4 | - | - | 73.3 | - | - | 62.7 | - | - | 62.0 | - |
| ***CLIP-based Vision Language Models*** | | | | | | | | | | | | | | | | | |
| MaskCLIP (Zhou et al., 2022) | Attn Adaptation | 12.3 | 16.9 | 26.2 | 25.6 | 62.9 | × | × | × | × | × | × | × | × | × | × | × |
| CLIPSurgery (Li et al., 2025b) | Consistent Attn | 16.1 | 21.9 | 29.3 | 31.4 | 77.5 | × | × | × | × | × | × | × | × | × | × | × |
| CASS (Kim et al., 2025) | VFM Graph | 20.4 | 26.7 | 40.2 | **39.4** | 87.8 | × | × | × | × | × | × | × | × | × | × | × |
| SAN (ViT-L) (Xu et al., 2023) | Decoupled Head | **32.1** | **45.8** | **57.7** | - | **94.6** | × | × | × | × | × | × | × | × | × | × | × |
| ***Vision Generalist Models*** | | | | | | | | | | | | | | | | | |
| X-Decoder (DaViT-d5) (Zou et al., 2023) | X-Decoder | 58.1 | - | 60.4 | 81.7 | 97.7 | × | × | × | - | - | - | - | - | - | - | - |
| 4M (ViT-B) (Mizrahi et al., 2023) | Task-specific Heads | 53.4 | - | - | - | - | 94.4 | - | - | × | × | × | × | × | × | × | × |
| SEEM (DaViT-d5) (Mizrahi et al., 2023) | SEEM-Decoder | - | - | - | - | - | × | × | × | - | - | - | - | - | - | 65.6 | - |
| BEIT-3 (Multiway-T) (Wang et al., 2023a) | None | **62.8** | - | - | - | - | × | × | × | × | × | × | × | × | × | × | × |
| GiT (Wang et al., 2024) | Parallel Decoding | 47.8 | **49.1** | **63.3** | 61.8 | - | × | × | × | × | × | × | × | × | × | × | × |
| SAM3 (Carion et al., 2025) | DETR-like Decoder | 13.8 | - | 60.8 | 65.2 | - | - | - | - | 75.5 | 77.6 | 71.0 | 67.3 | 71.1 | 63.4 | 73.4 | 74.0 |
| ***Multimodal LLM with Additions*** | | | | | | | | | | | | | | | | | |
| GLaMM (Vicuna-7B) (Rasheed et al., 2024) | SAM/Pixel Decoder | × | × | × | × | × | × | × | × | 79.5 | **83.2** | 76.9 | 72.6 | 78.7 | 64.6 | 74.2 | 74.9 |
| UniPixel (Qwne2.5-VL-3B) (Rasheed et al., 2024) | SAM Decoder | × | × | × | × | × | × | × | × | **80.5** | 82.6 | 76.9 | 74.3 | 78.9 | 68.4 | **76.3** | **77.0** |
| VisionLLM-v2 (Swin-T) (Wu et al., 2024) | Deform-DETR | 52.3 | - | - | - | - | x | x | x | 76.6 | 79.3 | 74.3 | 64.5 | 69.8 | 61.5 | 70.7 | 71.2 |
| UFO (InternVL2.5-8B) (Tang et al., 2026) | Mask Retrieval | **54.5** | **30.2** | - | - | - | **93.6** | - | - | 80.0 | 81.6 | **78.1** | 76.7 | 79.9 | 72.3 | 75.5 | 76.3 |
| ***Standard Multimodal LLM*** | | | | | | | | | | | | | | | | | |
| InternVL-3.5 (4B) (Wang et al., 2025) | None | × | × | × | × | × | × | × | × | × | × | × | × | × | × | × | × |
| Qwen3-VL (4B) (Bai et al., 2025a) | None | × | × | × | × | × | × | × | × | × | × | × | × | × | × | × | × |
| DepthLM (Qwne2.5-VL-3B) (Cai et al., 2025) | None | × | × | × | × | × | 86.8 | 74.7 | - | × | × | × | × | × | × | × | × |
| VistaLLM (Vicuna-7B) (Pramanick et al., 2024) | None | × | × | × | × | × | × | × | × | 74.5 | 76.0 | 72.7 | 69.1 | 73.7 | 64.0 | 69.0 | 70.9 |
| **DenseMLLM (4B, Ours)** | None | **54.2** | **52.2** | 60.4 | 70.4 | 92.5 | 90.4 | 87.6 | 92.7 | 80.7 | 82.0 | 78.4 | 76.2 | 79.6 | 71.4 | 76.5 | 76.6 |

ios, prompts for ADE20k are randomly sampled from a diverse set rather than using a fixed instruction. For referring segmentation, we evaluate on the 8 splits of RefCOCO (Kazemzadeh et al., 2014) using cumulative IoU (cIoU). We utilize the grounding ability of our model to get a box and crop with a random color box, applying $1.2\times$ padding, and resizing the short edge to 1,280 pixels, using the prompt "Segment the core target." For depth estimation, test sets include NYUv2 (indoor) (Silberman et al., 2012), DDAD (outdoor) (Guizilini et al., 2020), and Cityscapes (fine-tuning comparison) (Cordts et al., 2016), reporting the $\delta_1$ metric. The text prompt precedes the image and specifies the prompt: "Please estimate the depth of this image from the $B$ dataset.", where $B$ is the benchmark name. Specific details and prompts are provided in Appendix D.

## 4.2. Results

The results presented in Table 1 demonstrate the superior performance of our proposed DenseMLLM across multiple dense prediction benchmarks, highlighting its capability as a robust vision-language generalist without requiring complex architectural additions.

**Semantic Segmentation.** In the task of semantic segmentation, DenseMLLM achieves remarkable results across all five datasets. Notably, it attains 54.2 mIoU on ADE20K, compared with 47.8 mIoU of the vision generalist models like GiT (Wang et al., 2024) and 32.1 mIoU of the CLIP-based methods such as SAN (Xu et al., 2023). Crucially, distinct from standard Multimodal LLMs that are typically unable to perform these tasks (indicated by "×"), DenseMLLM demonstrates a unique capability to handle fine-grained dense prediction directly.

**Depth Estimation.** In the depth estimation tasks, DenseMLLM shows competitive results. It achieves 90.4 on NYUv2, while the result of DepthLM (Cai et al., 2025) is 87.6. Our method just needs to process the image once, while DepthLM needs to draw each point and inference multiple times. We also fine-tune the model like the vision specialist model SwinMTL (Taghavi et al., 2024) and achieve a 92.7 $\delta_1$ (vs. 92.1 of SwinMTL). While the specialist model UniDepth-v2 (Piccinelli et al., 2025) performs higher on NYUv2, the performance on DDAD is quite close (88.2 vs. 87.6), suggesting this general-purpose model sometimes meets the high performance of task-specific models.

**Referring Segmentation.** DenseMLLM demonstrates strong performance in the referring expression segmentation task. On the RefCOCO val, DenseMLLM achieves cIoU of 80.7, while the baseline results are 79.5 of

*Table 2.* **Comparison of 4B-level general-purpose MLLMs across diverse benchmarks.** The best results are highlighted in **bold**.

| Model | | General VQA | | | Reasoning & Math | | | Hallucination & Real-world | | | | OCR & Chart | | | Agent |
|---|---|---|---|---|---|---|---|---|---|---|---|---|---|---|---|
| | MMB | MMStar | MME | MMVet | M-Vista | M-Verse | AI2D | Hallu | POPE | RW-QA | TextVQA | DocVQA | ChartQA | OCRBench | S-Spot |
| InternVL-3.5-4B | 80.3 | 65.0 | 2272 | - | **77.1** | 45.8 | 82.6 | 44.8 | 88.9 | 66.3 | 77.9 | 92.4 | **86.0** | 822 | - |
| Qwen3-VL-4B | **83.9** | 69.8 | 2309 | **68.3** | 73.7 | 46.8 | 84.1 | 57.6 | **89.3** | 70.9 | **80.8** | **95.3** | 84.6 | **881** | 59.5 |
| DenseMLLM-4B | **83.9** | **71.1** | **2384** | 64.6 | 76.5 | **56.5** | **85.6** | **59.1** | 86.4 | **74.6** | 79.6 | 94.4 | 85.3 | 813 | **59.6** |

GLaMM (Rasheed et al., 2024) and 80.5 for UniPixel (Liu et al., 2026), which requires an extra SAM decoder. Besides, the method based on the polygon prediction is merely 74.5. The UFO (Tang et al., 2026) achieves a cIoU of 80.0, slightly lower than the 80.7 achieved by DenseMLLM. However, the UFO method requires additional mask token embeddings for the retrieval process. Compared with the above methods. our method is straightforward yet highly effective, designed to be applicable to standard VLMs without requiring extra architectural modifications.

**General Capabilities Results.** We evaluate DenseMLLM as a general-purpose foundation model against state-of-the-art 4B baselines (InternVL-3.5 (Wang et al., 2025), Qwen3-VL (Bai et al., 2025a)). As shown in Table 2, incorporating dense prediction capabilities does not compromise general proficiency; rather, DenseMLLM achieves competitive performance across 5 kinds of general capacities, involving 15 popular benchmarks. These results confirm DenseMLLM is a robust general MLLM that effectively unifies high-level understanding with dense predictions.

*Table 3.* **Ablation Study**. "Indiv. Mean" indicates NTP-M without relevant negative sampling(Rel. Sampling), trained from ADE20k (mIoU) without other vision data from stage I. "Data Scale" involves other vision data after Stage III and "RL" means the reinforcement learning stage IV (result in Table 1). Extra fine-tuning from our final model can further improve the performance.

| Method | BCE | Indiv. Mean | Rel. Sample | Data Scale | RL | Fine-tune | Results |
|---|---|---|---|---|---|---|---|
| Base (BCE) | ✓ | | | | | | 16.7 |
| + Indiv. Mean | ✓ | ✓ | | | | | 32.7 |
| + Rel. Sampling | ✓ | ✓ | ✓ | | | | 51.2 |
| + Data Scale | ✓ | ✓ | ✓ | ✓ | | | 52.3 |
| + RL | ✓ | ✓ | ✓ | ✓ | ✓ | | 54.2 |
| + Extra Fine-tune | ✓ | ✓ | ✓ | ✓ | ✓ | ✓ | 55.2 |

**Ablation Study for Stages and Methods.** Table 3 presents the ablation study to investigate the contribution of each component in our proposed framework. The baseline model, trained solely with BCE loss, yields a performance of 16.7. Introducing the "Indiv. Mean" strategy significantly improves the score to 32.7. The addition of "Relevant Samplin" provides the most substantial performance boost, jumping to 51.2, which demonstrates the critical importance of our negative sampling strategy. Furthermore, incorporating larger-scale vision data and the Reinforcement Learning (RL) stage steadily increases the result to 52.3 and 54.2, respectively. Finally, applying extra fine-tuning allows the model to achieve its peak performance of 55.2. These results confirm that every module contributes positively to the

final architecture. Moreover, DenseMLLM is a powerful foundation model to fine-tune downstream tasks.

*Table 4.* **Effectiveness Analysis.** Comparison of different loss functions and training objectives. The results demonstrate the superiority of our proposed method in handling multi-label dense prediction tasks compared to standard NTP and hard mining strategies. Results from ADE20k (mIoU) is trained from stage I.

| | Raw | Focal Loss | OHEM | Balanced BCE | Ours |
|---|---|---|---|---|---|
| NTP | 35.33 | 28.84 | 34.72 | – | – |
| NTP-M | 16.72 | 5.35 | 27.71 | 33.17 | **51.21** |

**Effectiveness Analysis.** As reported in Table 4, we realize the NTP supervision using the vocabulary ID tasks the most region of this image patch. Results suggest that the standard NTP (CE) is unsuitable for dense prediction. Since a single vision token often encompasses multiple semantic entities, the single-label constraint of NTP introduces inherent noise. Consequently, hard mining strategies like Focal Loss (Lin et al., 2017) and OHEM (Shrivastava et al., 2016) exacerbate this issue by overfitting to these noisy labels, resulting in performance degradation (e.g., 28.84% with Focal Loss). In contrast, our method utilizing *relevant negative sampling* significantly outperforms Balanced BCE (51.21 vs. 33.17). This suggests that for large-vocabulary multimodal models, our sampling strategy is more effective than simple balancing in managing the vast search space and class imbalance.

*Table 5.* **Size Scaling**. "Downsample" refers to the ratio of the vision token size to the pixel size of the ground truth image. This ratio changes as the input resolution increases, reflecting the impact of resolution scaling.

| Downsample | 1/32 | 1.5/32 | 2/32 | 2.5/32 | 3/32 | 3.5/32 | 4/32 |
|---|---|---|---|---|---|---|---|
| NTP | 32.6 | 33.1 | 34.6 | 35.5 | 35.7 | **35.9** | 35.3 |
| NTP-M | 45.0 | 46.9 | 48.6 | 49.5 | 49.4 | 50.6 | **51.2** |

**Size Analysis.** We follow standard MLLM practices by preserving the native aspect ratio. Besides, we perform input upscaling during inference, following the common practice. As analyzed in Table 5, NTP-M consistently outperforms standard NTP across all input scales. Notably, our method demonstrates superior test time scaling laws, where increasing the size leads to steady performance gains.

**Applicability Analysis**. As shown in Table 6, our method is also applicable to other large multi-model predictive models, suggesting the proposed method is a general technique. Thanks to the abundance of vision data and the pre-trained

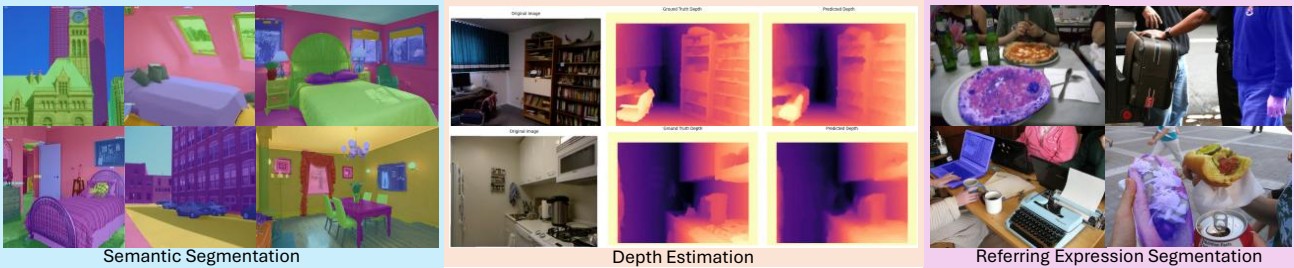

*Figure 3.* **Qualitative Results**. Visualization of three dense prediction tasks (semantic segmentation from ADE20k, depth estimation from the NYUv2, and referring expression segmentation from the RefCOCO).

DenseMLLM, it serves as a more effective vision-centric foundation model, enabling enhanced performance and versatility across various tasks (e.g., 51.2 vs. 44.9 on ADE20k).

*Table 6.* **Applicability Analysis**. Comparison with Qwen2.5VL-3B (Bai et al., 2025b) on ADE20k and Cityscapes benchmarks.

| Base Model | ADE20k (mIoU) | Cityscapes ($\delta_1$) |
|---|---|---|
| Qwen2.5VL-3B | 44.91 | 90.13 |
| Ours | **51.21** | **92.10** |

**Hyperparameter Study.** The hyperparameter study in Table 7 confirms the stability of our negative sampling method. Across different values of negative top k, the performance remained relatively consistent. The optimal accuracy of 51.21 was achieved at k = 32, while variations in k did not lead to significant fluctuations in results. This indicates that our negative sampling approach is robust, demonstrating reliable performance across a range of hyperparameters.

*Table 7.* **Hyperparameter Study.** Our relevant negative sampling is robust to the hyperparameter on the ADE20K (stage I).

| Negative Top k | 8 | 16 | 32 | 64 | 128 |
|---|---|---|---|---|---|
| Results | 50.79 | 51.10 | **51.21** | 50.06 | 49.93 |

*Table 8.* **Extra Inference Cost Comparison**. Our method does not involve predicting additional task embedding tokens, performing retrieval processes, or conducting inference with multiple noted images. $n$ is the class number and $m$ is the points number.

| Method | Task Tokens | Extra Inference |
|---|---|---|
| UFO (Tang et al., 2026) | $16 \times n$ | $n$ (only mask retrieval) |
| DepthLM (Cai et al., 2025) | 0 | $m$ (e.g. 1000 whole inference) |
| Ours | **0** | **0** |

**Extra Inference Analysis.** Considering the differences in model parameters, devices, and code bases, we perform a qualitative comparison of the extra inference cost as shown in Table 8, rather than a specific time comparison. Compared to prior methods, our approach eliminates both task-specific embedding tokens and any form of extra inference.

UFO requires 16 times class number task tokens and performs light mask retrieval per class, while DepthLM (Cai et al., 2025) avoids task tokens but still necessitates full model inference for each of the $m$ query points drawn on the image (e.g., 1000 times). In contrast, our method avoids additional tokens and repeated inference, making it easier to deploy. For the specific inference time, direct cross-method comparisons are often infeasible due to hardware and architectural discrepancies. Therefore, we benchmark variants within our framework at a baseline $500 \times 500$ resolution. Our method achieves a total runtime of $\approx 1\,\text{s}$ (logits operation $< 1\,\text{ms}$) with an average token latency of $\approx 5\,\text{ms}$. Under identical assumptions, DepthLM requires $100\,\text{s}$ to predict 100 points, while UFO introduces an extra $0.8\,\text{s}$ for 10 classes ($+80\,\text{ms}$ per class and $0.17\,\text{ms}$ retrieval). Our approach saves $0.8\,\text{s}$ and $99\,\text{s}$ against UFO and DepthLM, respectively, validating the efficiency of eliminating redundant inference repeats and task tokens. Note that scaling the resolution or incorporating intensive post-processing will naturally alter this latency.

**Visualization.** As shown in Fig. 3, DenseMLLM is capable of producing high-quality dense prediction results. This includes semantically precise segmentation outcomes, depth estimation results that closely resemble the ground truth, and accurate segmentations referenced by language. These results demonstrate that standard MLLM can inherently generate high-quality dense predictions. Detailed conversations are given in the Appendix D. Besides, we illustrate more visualization on the hidden states in the Appendix A

## 5. Conclusion

In this work, we presented DenseMLLM, a standard multimodal large language model that unifies dense prediction and general vision-language understanding without external decoders or task-specific tokens. By directly predicting pixel-level outputs from vision tokens and applying the multi-label next-token prediction loss, DenseMLLM achieves state-of-the-art performance in semantic segmentation and depth estimation while maintaining strong results

on standard VL tasks like VQA and referring expression segmentation. Our model demonstrates that architectural simplicity can coexist with high-quality dense perception.

Despite its strengths, DenseMLLM has limitations. Although performance is robust on standard benchmarks, the model's capability in extremely rare, long-tail open-world scenarios is currently constrained by the diversity of available public training data. Future work could explore incorporating larger-scale synthetic data or active learning to mitigate this. Moreover, our current framework focuses on semantic-level tasks; extending it to instance segmentation, panoptic segmentation, or other dense prediction problems, such as surface normals or optical flow, requires new mechanisms for instance discrimination or geometric reasoning. These directions offer promising paths toward a universal vision-language foundation model capable of both high-level understanding and pixel-accurate perception.

## Impact Statement

This paper introduces DenseMLLM, a framework that advances the unification of dense visual perception and general vision-language understanding within a standard MLLM. By eliminating the need for task-specific decoders and embeddings, our work enhances the efficiency and accessibility of versatile AI systems, potentially benefiting downstream applications such as robotic navigation, autonomous systems, and assistive technologies for the visually impaired. However, since dense prediction tasks like depth estimation and semantic segmentation are frequently employed in safety-critical domains (e.g., autonomous driving or medical imaging), we emphasize that this model should undergo rigorous domain-specific validation before deployment to mitigate risks from potential prediction errors.

## Conflicts of Interest

This work was conducted entirely during the internship of Y. Li, H. Shen, and L. Tang at Tencent. The research, development, and experimentation were carried out as part of the Youtu-VL project within Tencent Youtu Lab, representing the dense prediction capabilities of the model.

## Acknowledgments

This work was supported by grants from the Research Grants Council of the Hong Kong Special Administrative Region (Project Reference Nos. T45-401/22-N, N_HKUST654/24, and R6005-24) and the National Natural Science Foundation of China (Grant No. 62306254). The authors would also like to express their sincere gratitude to all members of the Tencent Youtu-VL team for their invaluable support, which provided an indispensable foundation for this work.

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

# A. Visual Representation Analysis

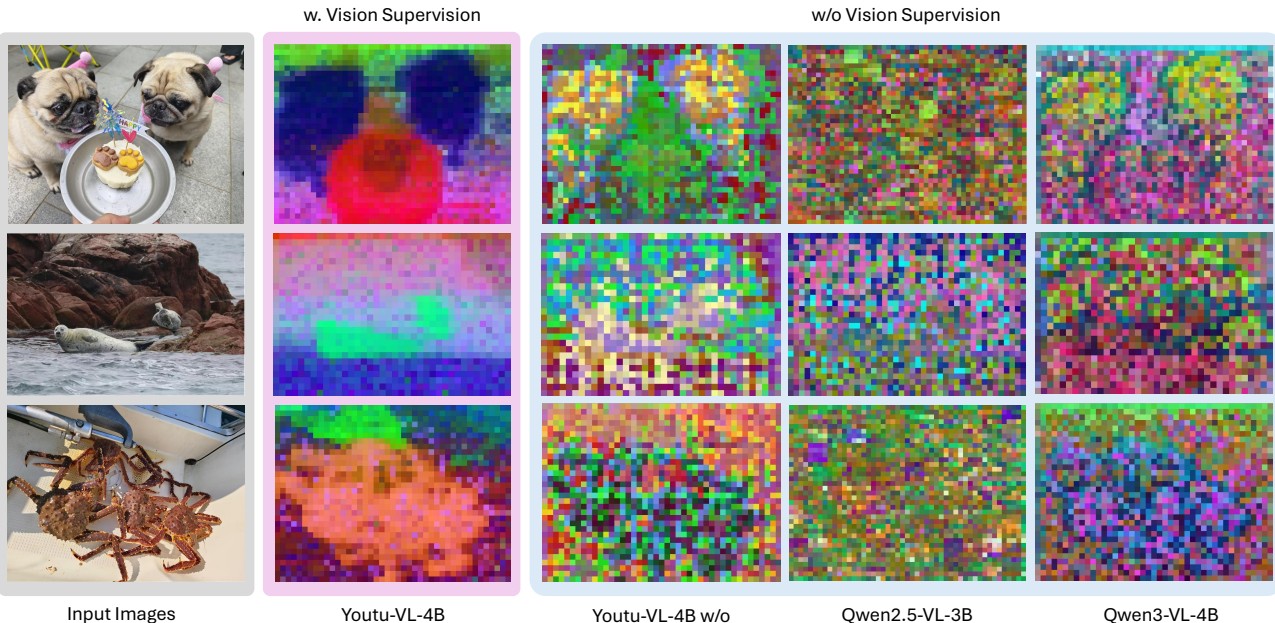

w. Vision Supervision          w/o Vision Supervision

Input Images          Youtu-VL-4B          Youtu-VL-4B w/o          Qwen2.5-VL-3B          Qwen3-VL-4B

*Figure 4.* **Visualization of Vision Token Representations**. This figure compares the Principal Component Analysis (PCA) visualizations (Oquab et al., 2024) of the last-layer hidden states of vision tokens. We contrast DenseMLLM-4B (with vision token supervision) against three models without such supervision: DenseMLLM-4B w/o, Qwen2.5-VL (Yang et al., 2024a), and Qwen3-VL (Bai et al., 2025a). Leveraging vision token supervision, our model exhibits outstanding feature separation and visualization quality compared to VLMs lacking this supervision.

To evaluate the quality of visual representations, we performed PCA on the last-layer hidden states of the vision tokens output by the LLM and projected them back onto the spatial dimensions of the original images, as shown in Figure 4. For VLMs lacking vision token supervision, we observe a clear progression from Qwen2.5-VL to Qwen3-VL; the latter exhibits superior representation quality with greater distinctiveness between objects, which positively correlates with its improved model performance. Notably, our DenseMLLM-4B (without vision supervision) demonstrates representation capabilities comparable to Qwen3-VL. However, upon integrating vision token supervision, the visual representations of DenseMLLM-4B show significant improvement, characterized by clear semantic structures and sharp object separation. This qualitative result provides compelling evidence for the effectiveness and necessity of our proposed vision supervision, especially in the visual representation aspect.

# B. Detailed Training Strategy

The training of our model follows a progressive, four-stage pipeline designed to establish a robust multimodal foundation, adapt to dense prediction tasks, and align with human preferences. The detailed configuration for each stage is described below and summarized in Table 9. Please note that the delineation of training stages in this paper differs slightly from the technical report; specific pre-training stages have been consolidated to streamline the presentation.

**Stage I: Multimodal Foundation Pre-training.** This stage integrates the construction of the language backbone with large-scale multimodal alignment.

- **Language Backbone:** Initially, we utilize a mixture of commonsense, STEM, and coding data to train the language backbone (corresponding to the first 10T tokens). The learning rate adopts a cosine learning rate strategy, starting from $4 \times 10^{-4}$.

- **Multimodal Alignment:** We subsequently introduce visual capabilities by training on approximately 1.8T tokens of mixed data (pure text, image-caption pairs, and vision-centric tasks). During this phase, all components (LLM backbone, Vision Encoder, Projector) are trainable. The learning rate decays to $4 \times 10^{-5}$ by the end of this stage (approx. 11.8T cumulative tokens).

**Stage II: Annealing Pre-training (Versatile Task Adaptation).** Corresponding to the specific annealing phase, this stage focuses on adapting the model for dense prediction and versatile tasks (e.g., Segmentation, Grounding, OCR, STEM).

- **Data Composition:** We utilize a high-quality mixture of approximately 0.6T tokens. As shown in the data recipe, the distribution is balanced with 40% Pure Text, 30% General Multimodal Data, and 30% Vision-Centric Data.

- **Hyperparameters:** The training continues from the previous stage with the sequence length maintained at 16K. The learning rate further anneals from $4 \times 10^{-5}$ down to $1 \times 10^{-5}$ to ensure fine-grained adaptation without catastrophic forgetting.

**Stage III: High-Quality Supervised Fine-Tuning (SFT).** This stage refines the model's instruction-following capabilities. We extend the context window from 16K to 32K tokens to support long-context understanding. The optimization uses the AdamW optimizer with a cosine scheduler, with the learning rate decaying from a peak of $2 \times 10^{-5}$ to a minimum of $2 \times 10^{-6}$.

**Stage IV: Reinforcement Learning (RL).** To further optimize the model and align it with specific rewards (e.g., class-label IoU for segmentation), we employ the DAPO approach.

- **Configuration:** The context length remains at 32k tokens. We use a fixed learning rate of $1 \times 10^{-6}$ for the policy network.

- **Stability:** To ensure stable dynamics, we remove the KL penalty term, adopt FP16 training, and set the clipping range to $[0.20, 0.24]$. A soft overlong punishment is introduced to regulate generation length.

*Table 9.* **Training setup and hyperparameters across the four sequential training stages**. Stage I combines language backbone and multimodal foundation training. Stage II focuses on annealing for dense tasks. Stages III and IV focus on alignment and refinement.

| Hyperparameters | Stage I | Stage II | Stage III | Stage IV |
|---|---|---|---|---|
| Description | Multimodal Foundation Pre-training | Annealing / Versatile Adaptation | Supervised Fine-Tuning (SFT) | Reinforcement Learning (RL) |
| Sequence length | 16K | 16K | 32K | 32K |
| Trainable components | All | All | All | Policy Model |
| Optimizer | AdamW | AdamW | AdamW | DAPO |
| LR Schedule | Cosine Decay | Cosine Decay | Cosine Decay | Fixed |
| Maximum LR | $4.00 \times 10^{-4}$ | $4.00 \times 10^{-5}$ | $2.00 \times 10^{-5}$ | $1.00 \times 10^{-6}$ |
| Minimum LR | $4.00 \times 10^{-5}$ | $1.00 \times 10^{-5}$ | $2.00 \times 10^{-6}$ | - |
| Precision | BF16 | BF16 | BF16 | FP16 |
| Training budget | ~11.8T tokens | ~0.6T tokens | - | 6,144 rollouts/iter |
| Data Mixture | Pure Text + General Multimodal | 40% Text, 30% General, 30% Vision-Centric | Instruct | Task-specific Rewards |

**Task-Specific Fine-Tuning.** Compared to the above main training phase, task-specific fine-tuning for real-world downstream applications requires significantly lower computational costs. For the ablation study, the model is fine-tuned based on Stage I (Table 7, Table 5, Table 4). We trained the ADE20k using only 8 GPUs for 5607 iterations (equivalent to 20 epochs with data augmentation). The employed optimizer is AdamW with a start learning rate of 1e-4 and a cosine learning rate scheduler. Besides the ablation study, our training strategy remains consistent for further fine-tuning in the last row of Table 3. The results is further improved after the task-specific fine-tuning. This demonstrates that our model can serve effectively as a foundational backbone for downstream tasks, allowing fine-tuning with minimal training overhead.

## C. Case Study

---

**Semantic Segmentation**

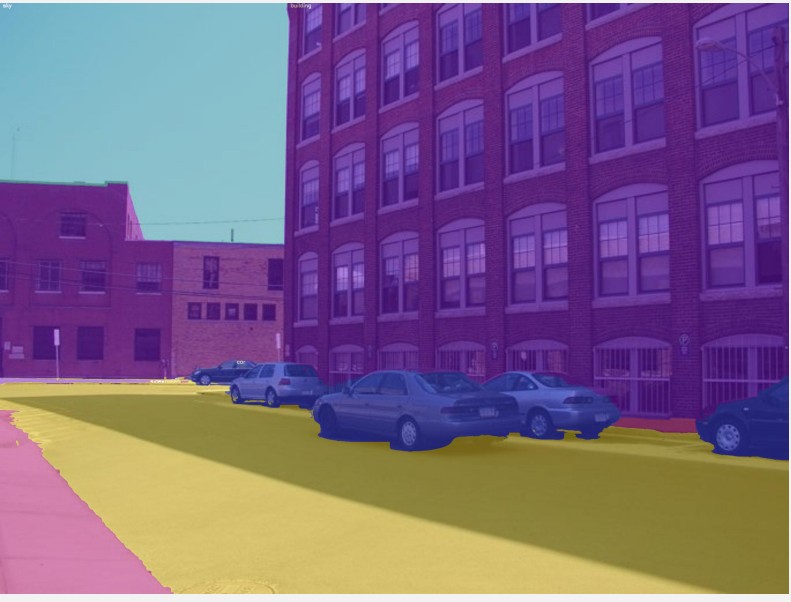

**User Instruction:** Create segmentation maps for these items: countertop, flower, hood, sky, shower, path, dishwasher, tank, skyscraper, railing, chest of drawers, arcade machine, painting, bar, building, sand, mountain, bannister, shelf, bus, bed , mirror, table, step, pier, refrigerator, floor, bag, minibike, conveyer belt, awning, canopy, runway, flag, book, basket, waterfall, water, car, apparel, bulletin board, pillow, signboard, washer, van, rug, plaything, chair, fountain, stairs, plate, animal, bottle, stove, door, blanket, stool, wall, buffet, trade name, fence, clock, box, dirt track, cushion, counter, radiator, seat, escalator, cradle, screen door, booth, ashcan, house, sofa, cabinet, coffee table, blind, fireplace, vase, swimming pool, bathtub, lamp, column, bicycle, toilet, hill, bridge, tower, ship, sea, pole, wardrobe, plant, tray, ceiling, windowpane, tree, television receiver, computer, light, oven, bench, grass, streetlight, monitor, river, bookcase, boat, rock, poster, person, food, curtain, sink, pool table, sidewalk, crt screen, earth, palm, lake, land, screen, stage, truck, desk, swivel chair, pot, traffic light, fan, chandelier, sculpture, ottoman, road, glass, hovel, base, towel, kitchen island, stairway, field, microwave, airplane, sconce, barrel, tent, case, grandstand, armchair, ball, without the background class.

**Model Response**

The target categories include `<ref>car</ref><ref>signboard</ref><ref>sidewalk</ref>` `<ref>sky</ref><ref>road</ref><ref>building</ref><ref>pole</ref><ref>` `streetlight</ref>`, numbered sequentially starting from 0, without the background class.`<mask>RLE string`

---

*Figure 5.* Qualitative example of semantic segmentation the ADE20k dataset.

**Semantic Segmentation**

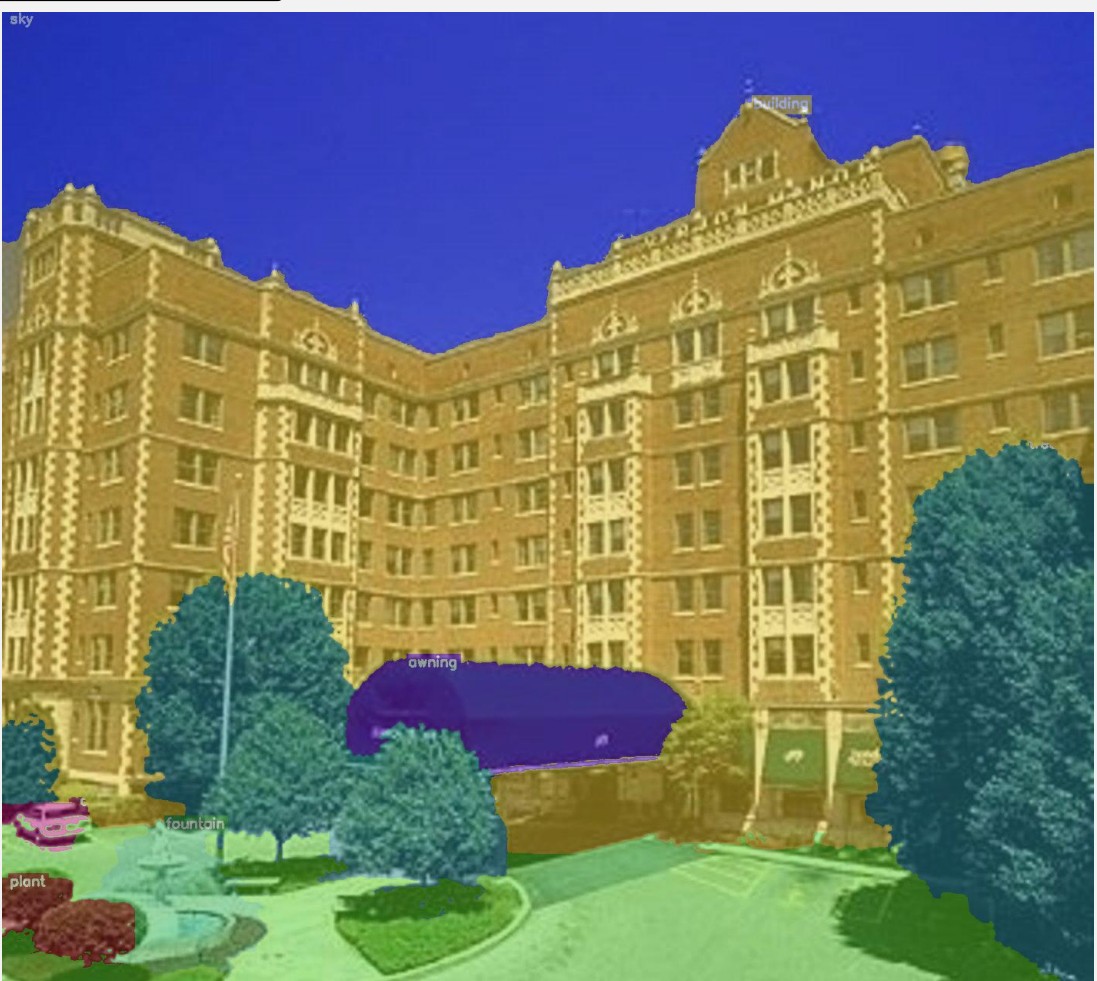

**User Instruction:** Label all river, tree, pot, tank, blind, bicycle, bannister, lake, screen, barrel, mirror, swimming pool, stairway, coffee table, bottle, bar, grass, plant, ball, fence, streetlight, sky, cushion, chair, microwave, stove, awning, rock, palm, sculpture, airplane, shower, conveyer belt, canopy, booth, radiator, flower, ashcan, fountain, house, bag, stage, minibike, screen door, escalator, oven, case, rug, plaything, poster, washer, pole, animal, swivel chair, ship, sand, bench, sidewalk, path, table, shelf, runway, counter, tower, bed , truck, basket, sofa, railing, crt screen, door, land, lamp, signboard, sconce, waterfall, ottoman, wardrobe, bus, countertop, kitchen island, column, hood, fan, glass, seat, base, stool, boat, hovel, bookcase, water, tent, traffic light, bathtub, tray, toilet, earth, sink, dishwasher, building, cabinet, pool table, apparel, painting, flag, trade name, armchair, sea, light, road, computer, monitor, desk, bridge, chandelier, fireplace, book, skyscraper, person, pillow, blanket, grandstand, windowpane, van, vase, towel, curtain, stairs, field, wall, hill, arcade machine, clock, pier, refrigerator, cradle, step, bulletin board, box, chest of drawers, mountain, food, television receiver, dirt track, plate, floor, buffet, ceiling, car in the image, without the background class.

**Model Response**

The target categories include
<ref>sky</ref><ref>road</ref><ref>car</ref><ref>tree</ref>
<ref>building</ref><ref>awning</ref><ref>fountain</ref><ref>plant</ref>,
numbered sequentially starting from 0, without the background class.<mask>RLE string

*Figure 6.* Qualitative example of semantic segmentation from the ADE20k dataset.

**Referring Segmentation**

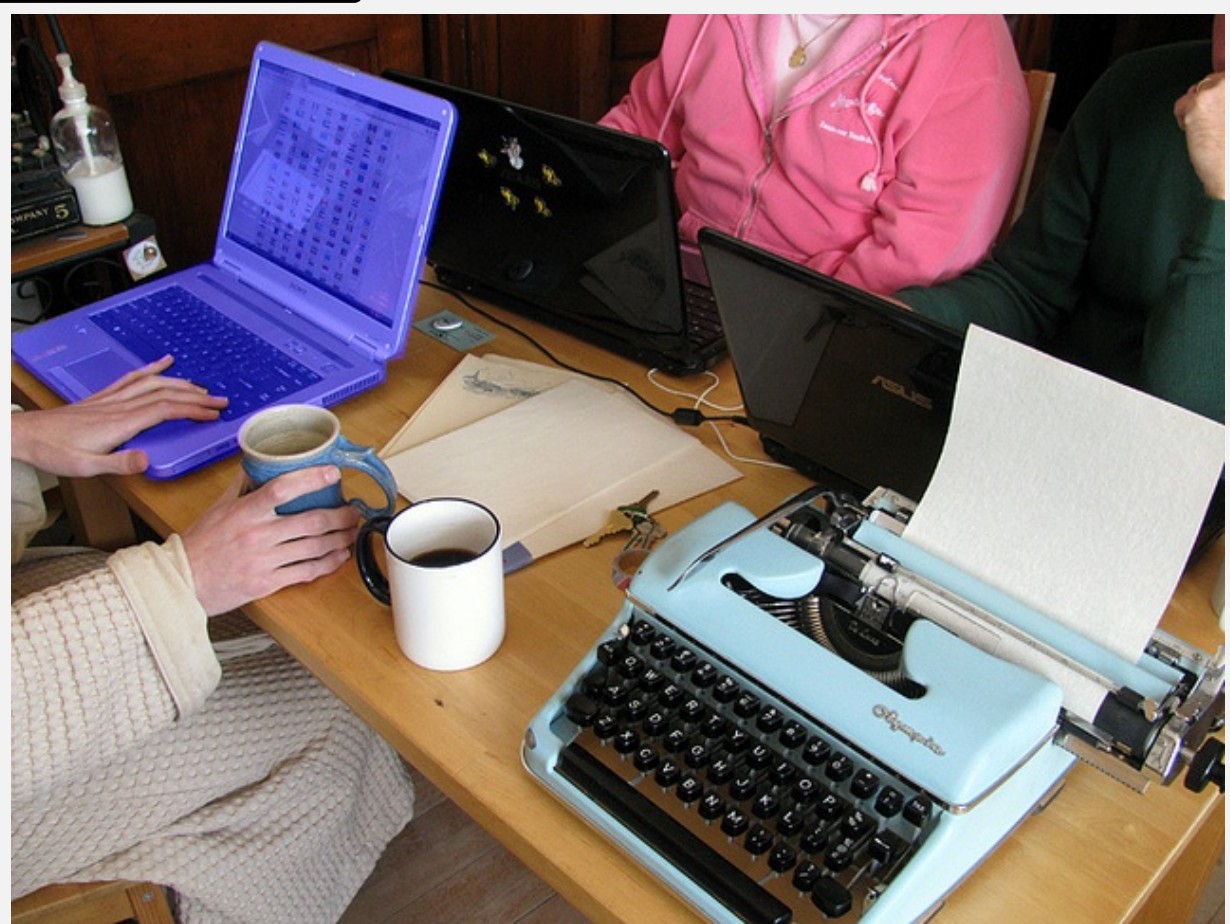

**User Instruction 1:** Please provide the bounding box coordinate of the region this sentence describes: the laptop left top

**Model Response 1**

```
<box><x_0><y_13><x_234><y_245></box>
```
(Operations: Draw box, Crop image, Resize)

**User Instruction 2:** Segment the core target.

**Model Response 2**

The results are 0 for `<ref><BG></ref>` and 1 for `<ref><FG></ref>`.`<mask>RLE string</mask>`

*Figure 7.* Qualitative example of referring expression segmentation from the RefCOCO dataset.

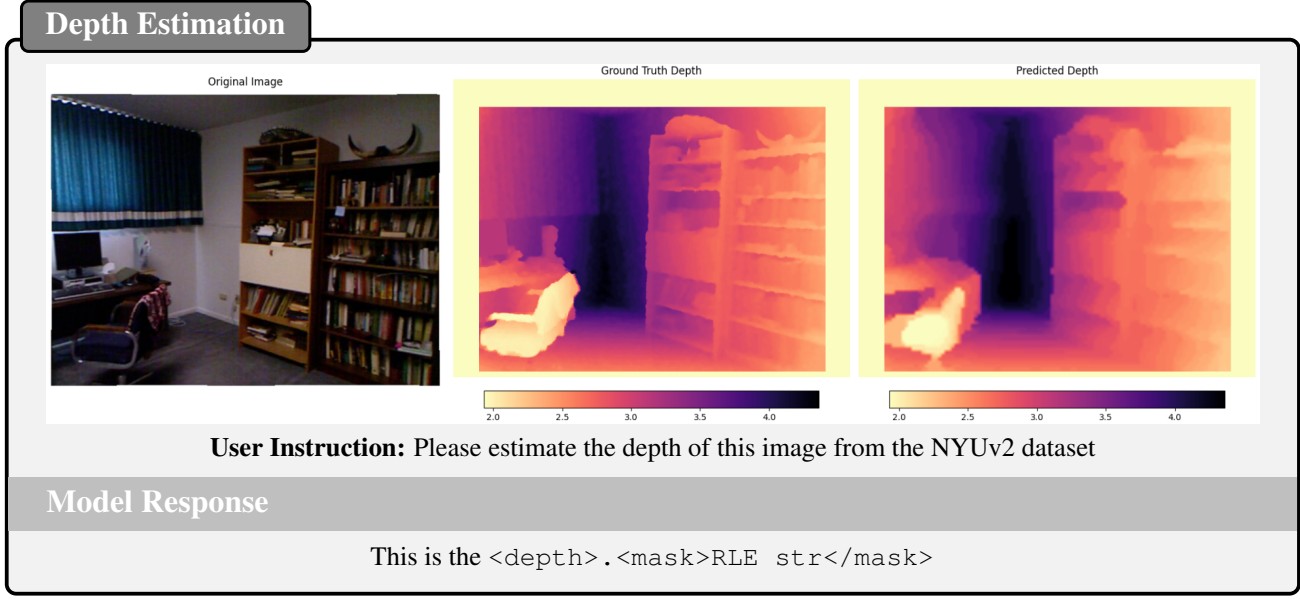

*Figure 8.* Qualitative example of depth estimation from the NYUv2 dataset.

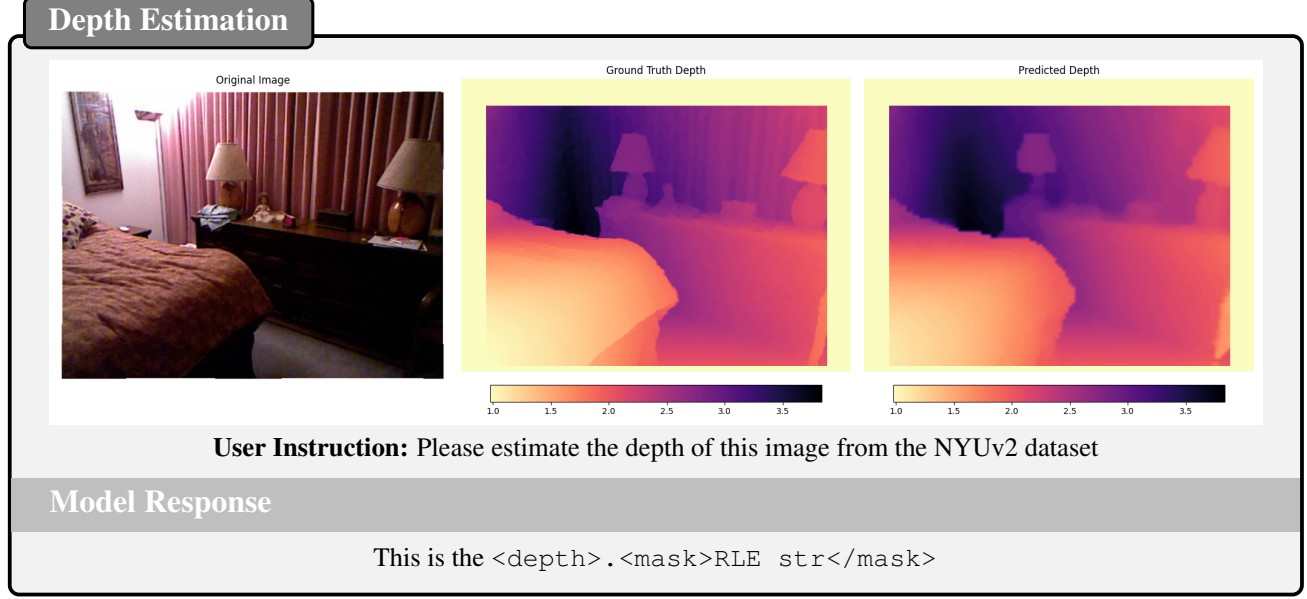

*Figure 9.* Qualitative example of depth estimation from the NYUv2 dataset.

**Failure Case**

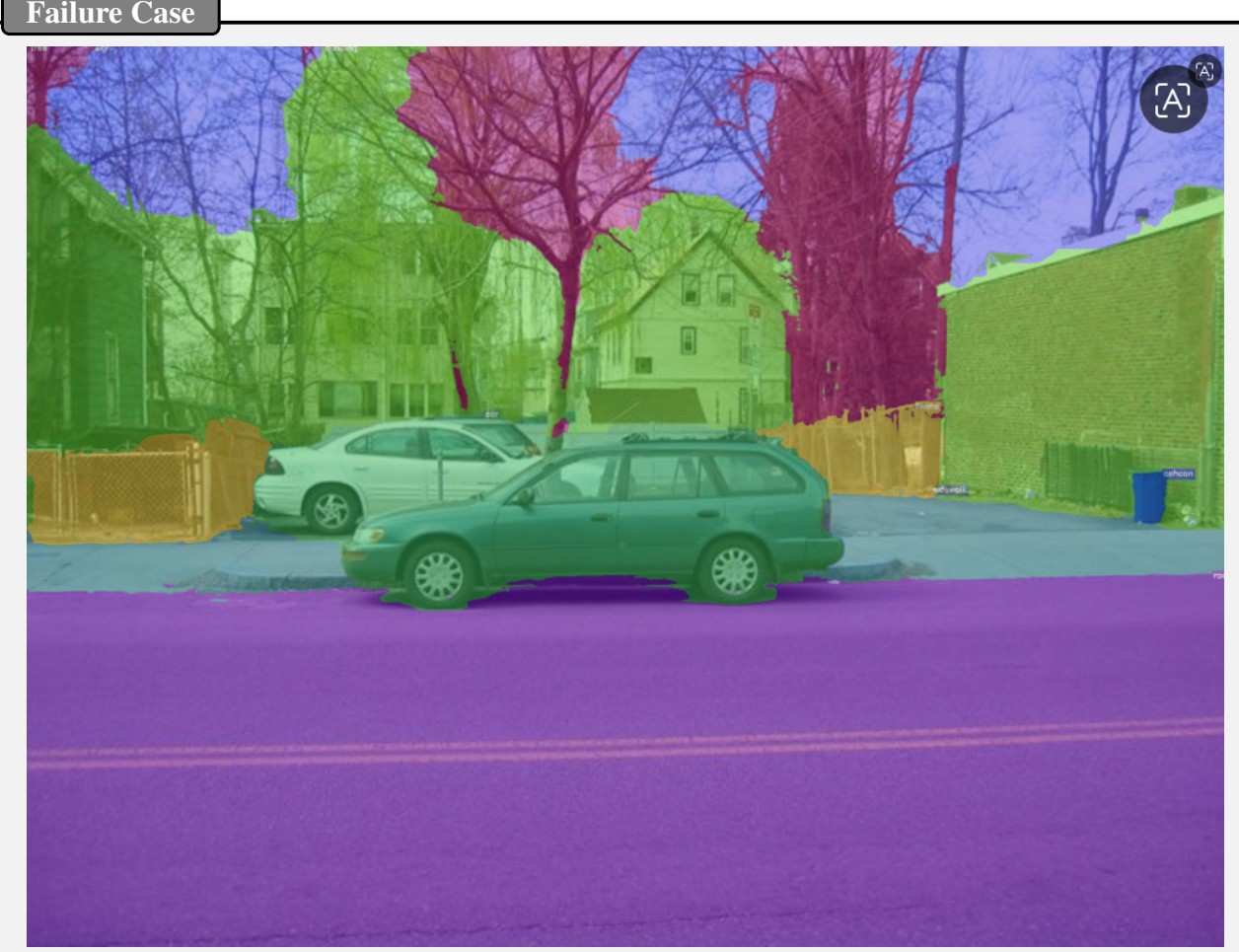

**Case Study:** Due to the constraints of patch size, our model exhibits limitations in scenarios with significant overlap (e.g., trees and sky). Additionally, without high-resolution input and post-processing, noticeable aliasing artifacts may occur.

*Figure 10.* Failure example of semantic segmentation from the ADE20k dataset.

# D. Evaluation Prompts

## D.1. Semantic Segmentation

**ADE20K (Zhou et al., 2017).** In our testing of the ADE20K dataset (test), we employed an open vocabulary method. The prompts were constructed using the format "Segment: class1, class2, class3, ..." where the class names are all lowercase English terms representing the various categories. To assist the model in querying each class individually without fitting to specific instructions, we randomly *shuffled* the class names for each image. Given that the patch size undergoes significant downsampling after token merging (to 32), and considering the small resolution of images in this benchmark, we scaled each image up by four times its original resolution for testing. The results for the detected categories are formatted as `<ref>class1</ref><ref>class2</ref><ref>class3</ref>`, making them easier to parse. The pixel-level outputs are listed at the end as `<mask>RLE string</mask>`, with the numbering corresponding to the order of the class names as they appear. "RLE string" indicates the pixel labels compressed by Run-Length Encoding. The length of the 1D mask string is the same as the pixel number. Users need to reshape it to the 2D raw image size for further usage. For evaluation without background, a softmax with temperature 0.2 and DenseCRF (Krähenbühl & Koltun, 2011) is applied to the logits to refine the pixel-level predictions. This operation is applied to all tasks related to semantic segmentation during evaluation, which is a common practice in segmentation methodsds. Here's a specific example:

```
<image>
Question:
Segment:  tree, towel, tank, armchair, refrigerator, countertop, blanket,
railing, hood, bathtub, radiator, chandelier, canopy, arcade machine, trade
name, case, wardrobe, oven, sidewalk, toilet, chair, wall, door, land,
fireplace, windowpane, plant, waterfall, swimming pool, computer, animal,
apparel, bannister, poster, flag, step, chest of drawers, box, book, light,
tower, ceiling, bulletin board, palm, bicycle, barrel, earth, buffet, person,
bus, stairs, stove, stool, basket, washer, rug, monitor, stairway, cushion,
pot, minibike, awning, desk, grass, clock, runway, sea, bookcase, sky, food,
hovel, seat, van, road, fountain, streetlight, coffee table, painting, ball,
car, bed , cradle, airplane, kitchen island, house, ottoman, column, rock,
blind, sconce, fan, escalator, lake, boat, booth, stage, sofa, base, swivel
chair, mirror, pole, shower, conveyer belt, table, bench, tray, glass, lamp,
dishwasher, crt screen, plaything, pool table, water, screen door, truck,
bridge, pillow, building, ship, signboard, traffic light, microwave, screen,
path, vase, mountain, pier, field, bottle, sculpture, floor, hill, river,
sand, flower, sink, grandstand, shelf, dirt track, ashcan, skyscraper, counter,
cabinet, television receiver, fence, bar, plate, tent, curtain, bag, without the
background class.
Answer:
The target categories include <ref>tree</ref><ref>sidewalk</ref><ref> ...
</ref><ref>pot</ref><ref>signboard</ref>, numbered sequentially starting from
0, without the background class.<mask>RLE string</mask>
```

Note that we evaluated ADE20k using a randomly sampled prompt from "Semantic Segmentation for the Open World" to demonstrate prompt flexibility, whereas other benchmarks used a fixed prompt for simplicity.

**Cityscapes (Cordts et al., 2016).** The evaluation of Cityscapes (val) is consistent with the output format of ADE20K; the only difference in the input prompt is the category name (shuffle the names, too). The distinction lies in the resolution, which is multiplied by 2 instead of 4. To reduce computation, we performed a 2x2 non-overlapping crop of the images for testing. The specific prompts are as follows:

```
<image>
Question:
Segment: train, motorcycle, vegetation, person, wall, terrain, pole, sky,
traffic light, fence, bicycle, road, traffic sign, rider, building, truck,
sidewalk, bus, car, without the background class.
Answer:
The target categories include <ref>road</ref><ref>sidewalk</ref><ref>car</ref>,
numbered sequentially starting from 0.<mask>RLE string</mask>
```

**Context59 (Mottaghi et al., 2014).** The Pascal context (val) benchmark follows the aforementioned prompt and output protocols, with the input resolution upscaled to $3\times$ its original size. Context59 comprises 59 foreground categories, excluding background classes. To accommodate scenarios requiring background identification, the model supports an optional configuration where the phrase "without the background class" is omitted from the prompt. In this mode, a background channel (represented by `<OTHERS>`) is integrated into the post-processing pipeline: logits are scaled by a factor of 0.25 before a sigmoid activation, followed by the addition of a constant background score (0.5) to facilitate the argmax operation. A representative prompt is illustrated below:

```
<image>
Question:
Segment: window, ceiling, dog, person, ground, keyboard, cloth, bus, bag,
boat, sheep, wall, bicycle, snow, platform, grass, flower, computer, floor,
truck, bottle, light, car, curtain, sign, bird, pottedplant, tree, cat, table,
door, bed, food, train, sidewalk, bench, bedclothes, sofa, mountain, rock,
water, building, aeroplane, plate, track, cabinet, horse, chair, cup, fence,
road, tvmonitor, motorbike, sky, book, mouse, cow, wood, shelves, without the
background class.
Answer:
The target categories include <ref>ground</ref><ref>grass</ref> ...
<ref>rock</ref><ref>water</ref><ref>sky</ref>, numbered sequentially starting
from 0.<mask>RLE string</mask>
```

**VOC20.** VOC20, or PASCAL VOC, contains 20 foreground classes without background classes. The test resolution multiples, prompt format, and output format are consistent with Context59. A specific example is as follows:

```
<image>
Question:
Segment: train, sofa, sheep, horse, bird, cat, car, cow, boat, aeroplane,
diningtable, pottedplant, chair, dog, bottle, person, motorbike, bicycle,
tvmonitor, bus, without the background class.
Answer:
The target categories include <ref>car</ref><ref>boat</ref><ref>person</ref>
<ref>bus</ref>, numbered sequentially starting from 0.<mask>RLE string</mask>
```

**COCOStuff (Caesar et al., 2018).** CocoStuff uses the same testing protocol as ADE20K, except that the resize scale is 3. The rest of the prompt format (including label shuffling) and output format are consistent. A specific example is as follows:

```
<image>
Question:
Segment: wall-tile, ground-other, surfboard, moss, fire hydrant, towel,
backpack, couch, floor-wood, mirror-stuff, rock, handbag, door-stuff, paper,
salad, gravel, door, plant-other, hill, snowboard, eye glasses, snow, person,
tennis racket, wall-concrete, playingfield, plastic, wall-wood, bird, banana,
carrot, bed, sandwich, furniture-other, knife, rug, plate, vase, elephant,
clouds, kite, tv, cell phone, fork, apple, straw, stone, frisbee, suitcase,
fence, cake, clock, train, grass, wood, donut, curtain, cow, cloth, floor-other,
toaster, tree, giraffe, sports ball, shelf, flower, building-other, potted
plant, carpet, fruit, window-blind, spoon, railroad, pillow, traffic light,
bush, desk-stuff, sheep, bear, railing, bottle, hat, blender, baseball glove,
sea, sky-other, hair drier, microwave, parking meter, zebra, tent, mouse, skis,
counter, desk, banner, tie, oven, branch, scissors, structural-other, dirt,
keyboard, fog, cupboard, wall-panel, wall-brick, floor-stone, food-other,
wall-other, dining table, napkin, stop sign, cat, net, mirror, leaves, bus,
house, bowl, mud, stairs, truck, laptop, table, textile-other, clothes, sand,
window, vegetable, pizza, floor-tile, wine glass, waterdrops, river, road,
floor-marble, cardboard, refrigerator, window-other, wall-stone, platform, cup,
mountain, street sign, shoe, umbrella, car, book, chair, skyscraper, cabinet,
skateboard, ceiling-other, blanket, toothbrush, bench, orange, light, hot dog,
bridge, pavement, bicycle, solid-other, ceiling-tile, teddy bear, water-other,
motorcycle, broccoli, horse, cage, mat, baseball bat, dog, roof, boat, metal,
sink, hair brush, remote, toilet, airplane, without the background class.
Answer:
The target categories include <ref>rock</ref><ref>plant-other</ref>
<ref>bird</ref><ref>grass</ref><ref>bush</ref><ref>bear</ref><ref>dirt</ref>
<ref>water-other</ref>, numbered sequentially starting from 0.<mask>RLE
string</mask>
```

**Semantic Segmentation for the Open World.** Open set segmentation and specific benchmarks differ in the output content and interpretation results. Open sets support single objects, multiple objects, and a collection of labels. If there is no improvement without the background, the model will output a label of <OTHERS>, which will activate the sigmoid + threshold interpretation method. Specifically, the logits are divided by 4 and then passed through sigmoid, with the background class threshold defaulting to 0.5. The corresponding accuracy for open sets is generally not high, and fixed thresholds can sometimes lead to inaccuracies in the background class. We can achieve more accurate open-set segmentation results using the detection-then-segmentation type. The specific approach is to first invoke detection, then draw each box on the image, apply a padding of 1.2, resize, and then segment the foreground and background. In contrast, directly outputting semantic segmentation results is faster but relatively lower in quality, though it can additionally support stuff classes. The specific prompts for direct semantic segmentation are as follows (the Chinese version is also supported):

```
Please perform segmentation for the following classes: {keyword}.
Can you identify and segment these objects: {keyword}?
I'm looking for {keyword}.  Please segment them in the image.
Segment the image based on these keywords: {keyword}.
Mark the regions corresponding to: {keyword} in the image.
Identify the boundaries of these objects: {keyword}.
Please label and segment the following categories: {keyword}.
Show me the segmentation mask for: {keyword}.
Create segmentation maps for these items: {keyword}.
Locate and segment the following objects: {keyword}.
Segment the image to highlight: {keyword}.
Distinguish and segment these classes: {keyword}.
Please provide segmentation for objects: {keyword}.
Identify and separate the following elements: {keyword}.
Generate segmentation for the specified classes: {keyword}.
Semantic segmentation for {keyword}.
Perform semantic segmentation for {keyword}.
Run seamntic segmentation: {keyword}.
Segment: {keyword}.
Semantic segmentation: {keyword}.
Find {keyword} and segment.
Segment these: {keyword}.
Segment {keyword}.
Segment {keyword}.
Segment and label {keyword}.
Please segment {keyword}.
Could you please segment objects classified as: {keyword}?
Provide a detailed segmentation of the following: {keyword}.
Perform pixel-level segmentation for: {keyword}.
Mark and segment the specified classes: {keyword}.
Generate a segmented output focusing on: {keyword}.
Show segmentation results highlighting: {keyword}.
Can you segment and annotate the following keywords: {keyword}?
Provide segmentation masks specifically for: {keyword}.
Please segment any visible {keyword} in the image.
Produce a segmentation mask isolating {keyword}.
Segment and highlight the regions occupied by {keyword}.
Perform semantic segmentation targeting: {keyword}.
Segment and classify the following entities: {keyword}.
Mark the presence and segment the {keyword} in the image.
Generate masks that segment the {keyword} clearly.
Create a detailed segmentation for the objects: {keyword}.
Generate segmentation annotations for: {keyword}.
Provide a detailed semantic segmentation of the following categories: {keyword}.
Perform pixel-level semantic segmentation for: {keyword}.
Create precise semantic segmentation boundaries around these classes: {keyword}.
Generate a semantic segmentation output focusing on: {keyword}.
Segment {keyword} in the image.
Show semantic segmentation for {keyword}.
Find and segment {keyword}.
Please segment {keyword}.
Segment {keyword} categories.
Show {keyword} segmentation.
Classify and segment {keyword}." }.
Generate segmentation annotations for: {keyword}.
```

{keyword} indicates some category names such as "dog, cat, tree, soft". We also support "segment anything" type without specific keyword assignment. It is important to note that this model does not necessarily segment all elements in detail, but rather segments some significant main objects. Since background classes will also be output, sigmoid processing will be applied, with a background threshold of 0.5. The combined version of detection and semantic segmentation can address issues with inaccurate backgrounds. Suggested prompts include:

```
Please segment all objects present in the image.
I want to segment everything in the image.
Segment every object visible in the photo.
Segment all objects in this image.
Segment all visible elements in the photo.
Segment all objects in the image.
Identify and segment all items.
Segment everything visible.
Find and segment all objects.
Label all objects in the image.
Segment all elements.
Perform semantic segmentation for this image.
Do semantic segmentation for this image.
Segment all.
Conduct semantic segmentation for all.
Perform semantic segmentation for this image.
Do semantic segmentation for this image.
Segment all.
Segment this image.
Segment it.
Segment everything.
Segment all
Semantic segmentation for all.
Semantic segmentation.
Run semantic segmentation
Segment this image
Conduct semantic segmentation for all.
Apply semantic segmentation to the image.
Run semantic segmentation on this image.
Execute semantic segmentation for all objects.
Carry out semantic segmentation for everything.
Perform full semantic segmentation.
Complete semantic segmentation for all items.
```

### D.2. Referring Expression Segmentation

**RefCOCO series (Yu et al., 2016) (polygon).** The basic version of referring expression segmentation is implemented using polygons. This mode is a simple version, but not evaluated.

```
<image>
Question:
Can you segment ''bowl behind the others can only see par'' in this image?
Answer:
The segmentation result is <poly><x_468><y_3><x_471><y_46><x_473><y_83><x_521>
<y_107><x_581><y_117><x_640><y_106><x_640><y_105><x_640><y_1></poly>.
```

This is the basic referring segmentation prompt. Note that `` represents an object, and an object may have multiple `<poly>`, indicating different parts.

**RefCOCO series (Yu et al., 2016) (grounding-then-segmentation).** Due to the precision limitations from the number of points, we recommend first using grounding and then semantic segmentation. Specifically, after grounding outputs bounding boxes on the original image, we draw boxes on the image with random colors and crop the image using a padding ratio of 1.2 (an extra 0.2). We then resize the shorter side to 1280 for semantic segmentation to distinguish between foreground and background classes (supporting lower resolutions for acceleration). "RLE string" in the output indicates the pixel labels compressed by Run-Length Encoding in the string format. The evaluation metric is cIoU. Examples based on Polygon and semantic segmentation are shown below:

In the testing, we actually used grounding prompts, then drew boxes on the image and performed a 1.2 padding crop. After setting the short side to 1280, we executed the following semantic segmentation commands, and then filled the segmentation results back into the original image for testing.

```
<image>
Question:
Can you segment ``Segment the code '' in this image?
Answer:
The segmentation result is <poly><x_468><y_3><x_471><y_46><x_473><y_83><x_521>
<y_107><x_581><y_117><x_640><y_106><x_640><y_105><x_640><y_1></poly>.
```

This is the basic referring segmentation prompt. Note that `` represents an object, and an object may have multiple `<poly>`, indicating different parts.

```
<image>
Question:
Please provide the bounding box coordinate of the region this sentence
describes:  the person bottom left
Answer:  <box><x_155><y_154><x_221><y_206></box>
```

```
<image>
Question:
Please provide the bounding box coordinate of the region this sentence
describes:  bowl behind the others can only see part Answer:
<box><x_54><y_0><x_361><y_141></box>
<cropped image> #Draw the box, padding, and resize.
Question:
Segment the core target.
Answer:
The results are 0 for <ref><BG></ref> and 1 for <ref><FG></ref>.<mask>RLE
string</mask>
```

Note that the mask result is from the cropped image. We fill it back to the raw image for evaluation via cIoU.

**Referring Expression Segmentation for the Open World.** Referring segmentation and points are the keywords of this task, which will activate the model to output polygon segmentation results for a single target. The approach based on grounding + semantic segmentation can refer to the flexible prompts of grounding.

```
Can you segment "{keyword}" via points?
Can you segment "{keyword}" in the manner of referring segmentation?
Referring segment for "{keyword}".
Referring segment for {keyword}.
Referring expression segmentation for {keyword}.'  Can you segment {keyword} via
points?
Can you segment {keyword} in the manner of referring segmentation?
Outline {keyword} via points.
Outline {keyword} via the polygon or points.
Segment {keyword} via points.
Outline "{keyword}" via points.
Use points to segment {keyword}.
Use points to segment "{keyword}".
Use point to segment {keyword}.
Use point to segment "{keyword}".
Please {keyword}.
Use points to segment "{keyword}".
Outline "{keyword}" via the polygon or points.
Segment "{keyword}" via points.
Perform referring segmentation for the keywords:  {keyword}.
Perform referring segmentation for "{keyword}".
Please segment "{keyword}" in the image using polygon or points.
Could you outline "{keyword}" with points?
Draw the boundary of "{keyword}" via points.
Generate a referring segmentation mask for "{keyword}".
Segment {keyword} based on referring expression.
Segment "{keyword}" using points outlining.
Please perform referring expression segmentation for {keyword}.
Use polygon or points to segment "{keyword}".
Draw a polygon or points to segment "{keyword}" in the image.
Referring segmentation for the object "{keyword}".
Segment {keyword} in this picture by outlining with polygon or points.
Please extract the polygon or pointsal region corresponding to "{keyword}
```

### D.3. Depth Estimation

**NYUv2 (Silberman et al., 2012).** The depth estimation test has four key points: (1) The category IDs are preset from ¡custom_1¿ to ¡custom_1000¿, eliminating the need to predict category names, which increases speed. (2) We first upsample by a factor of two, followed by argmax, and then resize to the original image size, without additional post-processing. Appropriate resizing before argmax improves results. (3) Real depth needs to be dequantized. For NYUv2, we use uniform linear quantization, mapping real depths from 0 to 10 meters to 1 to 1000. Invalid depths are set to 0 and ignored during training. During testing, we dequantize to the actual depths and exclude invalid depths. (4) The prompt must precede the image so that the model can learn the potential quantization methods. The output format is a string of depth names corresponding to the pixel sizes after argmax and resizing. For NYUv2, since the images are relatively small, we performed a threefold resize based on the original resolution. The example is given below:

```
Question:
Please estimate the depth of this image from the NYUv2 dataset.
<image>
Answer:
This is the <depth>.<mask>RLE string</mask>
```

**Cityscapes (Cordts et al., 2016).** The testing criteria for Cityscapes are the same as for NYUv2. The difference lies in quantizing the effective depth from 0-80m to 1-1000, and since the resolution is sufficient, no resizing was done. Besides, we set invalid regions from the left and bottom pars following previous works. Specific examples are as follows:

```
Question:
Please estimate the depth of this image from the Cityscapes dataset.
<image>
Answer:
This is the <depth>.<mask>RLE string</mask>
```

**DDAD (Guizilini et al., 2020).** The testing criteria for DDAD are the same as for NYUv2. The difference lies in quantizing the effective depth from 0.05-120m to 1-1000, and since the resolution is sufficient, no resizing was done. Specific examples are as follows:

```
Question:
Please estimate the depth of this image from the DDAD dataset.
<image>
Answer:
This is the <depth>.<mask>RLE string</mask>
```

**Depth Estimation for the Open World.** Depth estimation in open world scenes defaults to Log-uniform Quantization, mapping real depths ranging from 0.5 to 100m to 1 to 1000. Values outside this range are set to 0 (IGNORE). During testing, the predicted values need to be de-quantized to obtain the real depth; otherwise, only relative depth is obtained. Additionally, we require inputs to be resized to a focal length of 2000 pixels to simulate a camera with a default focal length. Discrepancies in actual focal length may lead to biased predicted depths, resulting in relative depth outputs. Unlike specific datasets that require prompts to be placed in advance, our open-set supports prompts both before and after the image. It is important to note that this task relies on a large amount of training data, and the capabilities of depth anything still need improvement. However, we have demonstrated that the standard VLM model can accommodate several different quantization methods, inherently building a spatial depth perception without requiring additional structures. We recommend using the following prompts and their Chinese translation versions:

```
Estimate the depth.
Estimate the depth in the default range.
Predict the depth map.
Estimate the depth map.
What the depth map?
Execute depth estimation.
Run depth estimation.
Generate a depth map.
Calculate the depth.
Provide depth estimation.
Perform depth prediction.
Create the depth map.
Get the depth map for the image.
Compute the depth information.
Estimate distance using depth.
Run a depth map generation.
Can you predict the depth map?
Show me the depth estimation.
Depth map prediction, please.
Extract depth from the image.
Apply depth estimation.
```

