# OpenReview forum: "DenseMLLM: Standard Multimodal LLMs for Dense Prediction"
_ICML.cc/2026/Conference — ICML 2026 regular_

### Official Review · Reviewer_qH4k · 2026-03-12

**Soundness:** 2
**Presentation:** 3
**Significance:** 2
**Originality:** 2
**Overall Recommendation:** 3
**Confidence:** 4

**Summary:**

This paper studies an important question: Whether a standard multimodal large language model can handle dense prediction tasks, including semantic segmentation, depth estimation, and referring segmentation, without introducing task-specific decoders. The core idea is to extend next-token prediction to vision tokens and impose multi-label supervision on them, so that a standard MLLM can directly produce dense outputs.

The paper reports competitive results on benchmarks, such as ADE20K, DDAD, and RefCOCO, and further argues that the model retains much of its general visual-language capability on standard VLM benchmarks.

**Compliance With Llm Reviewing Policy:**

Affirmed.

**Final Justification:**

Regarding that the methodological novelty is overstated and the wording should be largely changed, I still think this paper should not be accepted at this moment. The authors are encouraged to further refine the title and main body to make the descriptions appropriate.

**Key Questions For Authors:**

- Why should NTP-M be considered an extension of NTP rather than a multi-label BCE-style objective?  Is there a principled theoretical or empirical reason for this framing?
- I am also curious about the actual inference time of the proposed dense prediction pipeline. Although the method avoids an additional decoder, it would be helpful to report the runtime cost of performing dense prediction in practice, including any task-specific processing steps. This would clarify how efficient the approach is under realistic inference settings.

**Limitations:**

Discussed in the paper.

**Strengths And Weaknesses:**

Strengths:
- The paper addresses a meaningful and timely problem. Dense prediction has traditionally relied on specialized heads or decoders, and the attempt to absorb such capabilities into a unified standard MLLM framework is worthwhile. This direction is conceptually appealing because, if successful, it could simplify multimodal modeling and reduce the dependence on task-specific architectural design.

- The empirical results are nontrivial. The paper shows that, under a sufficiently strong training recipe, a standard MLLM can achieve competitive performance on multiple dense prediction benchmarks, including semantic segmentation, depth estimation, and referring segmentation. This at least provides useful evidence that dense perception is not fundamentally out of reach for standard MLLMs.

Weaknesses:
- The methodological novelty is considerably overstated. The paper introduces NTP-M as an extension of next-token prediction. However, under the formulation presented in the paper, the training objective for vision tokens is no longer standard autoregressive next-token prediction. Instead, it is effectively an independent Bernoulli-style multi-label prediction objective over the full label vocabulary, combined with top-k relevant negative sampling. This design is much closer to multi-label classification than to autoregressive NTP in the usual sense. The paper does not currently provide a convincing theoretical justification for treating this reformulation as a natural extension of NTP, nor does it clarify what fundamental advantage this framing offers over a more direct multi-label classification objective. In its current form, the paper does not establish a genuinely new learning principle.
- The claim of “zero extra inference overhead” appears overstated. Although the method avoids extra tokens, retrieval modules, and repeated full forward passes, the inference pipeline still includes additional task-specific steps such as active category selection, logit filtering, and optional zooming or post-processing. For RES, it further relies on grounding, cropping, padding, and resizing before segmentation. These steps may be lightweight, but they are not equivalent to a strictly single-pass prediction process. A more accurate claim would be that the method avoids an additional dedicated decoder and repeated full-model inference, rather than incurring literally zero extra cost.

---

> ### Author Rebuttal · Authors · 2026-03-29
>
> Thank you for your insightful comments. We appreciate the opportunity to clarify potential misunderstandings regarding NTP-M and our claim of “zero extra inference overhead.” Below we address each concern with a refined explanation.
>
> ---
>
> ### Q1: Justification of NTP-M vs. Multi-label Classification
>
> **Concern:** Why should NTP-M be considered an extension of NTP rather than a multi-label BCE-style objective; Theoretical or empirical justification; Clarification of fundamental advantage; New learning principle.
>
> **Response:**
> - **NTP-M vs. NTP beyond BCE**:
> (1) NTP-M strictly follows the autoregressive NTP paradigm: it **uses causal attention to predict the next token**, unchanged from standard practice, with the original LLM vocabulary space and the same hidden states.
> (2) The only difference is token modality: vision tokens naturally permit multiple valid next tokens (multi-label), unlike text tokens. We replace CE with BCE to handle this, but **BCE here is merely a loss function**, not a departure from autoregressive prediction.
> (3) Conventional BCE is non-autoregressive in MLLM, used in multi-label settings without causal attention.
> (4) Our contribution is not overstated: it provides a principled unification of text and vision tokens under the same NTP paradigm, revealing **fundamental differences between text and vision tokens in NTP**.
>
> - **Theoretical and Empirical Justification**: Theoretically, p(Y|X_v, X_instruct}) in Eq. 3 is consistent with NTP: both represent next-token prediction conditioned on prior instruction and vision tokens, the only difference being that a single text token ID is replaced by multiple visual token IDs. Empirically, our architecture is identical to NTP, using causal attention to realize sequential prediction.
>
> - **Fundamental Advantage**: As described in "Relevant Negative Sampling," NTP-M corrects the imbalance from MLLM’s large vocabulary to enable stable convergence. Table 4 ("Effectiveness Analysis") shows NTP-M raises performance from 16.72 to 51.21 and outperforms CE-based NTP (35.33), demonstrating its effectiveness.
>
> |        | Raw   | Focal Loss | OHEM  | Balanced BCE | Ours  |
> |--------|-------:|-----------:|------:|-------------:|------:|
> | NTP    | 35.33 | 28.84      | 34.72 | —           | —     |
> | NTP-M  | 16.72 | 5.35       | 27.71 | 33.17       | **51.21** |
>
> - **New Learning Principle**: We extend the NTP paradigm from text tokens to visual tokens and adapt the optimization objective to account for the single-label vs. multi-label distinction. From both supervised targets and implementation perspectives, this establishes a novel learning principle.
>
> ---
>
> ### Q2: Accurate Claim about "Zero Extra Inference Overhead"
> **Concern:** Additional task-specific steps; Single-pass prediction process of RES; Accurate claim about avoiding an additional dedicated decoder and repeated full-model inference.
>
> **Response:**
> - **Not Overstated**: The term “zero” appears only in line 439, where the original text states “zero overhead in both aspects, requiring no additional tokens and no repeated inference.” Our claim is strictly limited to “additional tokens” and “repeated inference”; we have not claimed anywhere that lightweight steps are unnecessary.
>
> - **Task-Specific Steps:** Category selection and logit filtering involve only a few lines of code, consistent with “without external decoders or complex decoding schemes.” Optional zooming/post-processing are unnecessary for depth estimation and used only selectively for other tasks.
>
> - **Single-pass RES**: Our model also supports single-pass RES by predicting polygon text, delivering performance comparable to VistaLLM but slightly below the default mode. Polygon prompts will be provided in a later release.
>
> - **Accurate Claim**: We will remove “zero” to avoid potential misunderstandings. However, our main claims in the paper regarding “without task-specific decoder / retrieval process / complex decoding schemes” remain accurate and rigorous.
>
> ---
>
> ### Q3: Inference Time
> **Concern:** Report the runtime cost of performing dense prediction in practice.
>
> **Response**: Thank you for raising the runtime question. As noted in Line 428, direct comparisons are infeasible due to differences in parameters, devices, resolutions, and code. We therefore benchmark against our own runtime.
>
> - At 500 resolution: **≈1 second** total (logits operation < 1ms); average latency per output token **≈5 ms** (varies with prompt/output length).
> - Under consistent assumptions (Table 8):
>   - DepthLM: **100 seconds** to predict 100 points.
>   - UFO: **+80 ms per output class** (16 extra tokens) + 0.17 ms retrieval, while category token and logit operations are common and thus omitted. For 10 classes, this adds **0.8 s** over our method.
> - Our method saves **0.8 s vs. UFO** and **99 s vs. DepthLM**, demonstrating the efficiency of eliminating extra inference repeats and task tokens.
>
> We will include these numbers in the final version.

---

> > ### Author Rebuttal · Reviewer_qH4k · 2026-04-02
> >
> > Thanks for the rebuttal. After reading the reviews from other reviewers and the reponses, I agree with Reviewer Lykw that the proposed method gains mostly from specialized supervision, large-scale data and training strategies, rather than being inherently possessed by standard MLLM. The title of this paper is actually not consistent with the content the main body describes. I think the authors should adjust the descriptions of this paper. So, I decide to keep the original rating unchanged.

---

> > > ### Author Response · Authors · 2026-04-02
> > >
> > > ### Response to Reviewer
> > >
> > > We thank the reviewer for the feedback and for acknowledging that our previous responses have addressed the technical concerns. We also note that Reviewer Lykw, who shared similar concerns, has stated in the rebuttal acknowledgement that their concerns are fully resolved. We appreciate the opportunity to clarify the intent of our title and the nature of our contributions. Below is our plan for revision and clarifications.
> > >
> > > ---
> > >
> > > **1. Innovation in Supervision vs. Data/Training**
> > >
> > > We agree that our performance gains stem from specialized supervision, but we maintain that this is our primary methodological contribution, rather than a trick. Besides, data scaling and training strategies are common processes in other general MLLMs with very limited gains compared to our core contribution. We would like to highlight the following points:
> > >
> > > *   **Methodological Innovation:** Extending supervision from text tokens to vision tokens while accommodating multi-label characteristics is a highly innovative shift instead of a trick to improve the performance. This was recognized by **Reviewer qH4k (Strong Accept)** as *"very interesting for the vision community and could be highly impactful."*
> > > *   **Method vs. Data (Ablation Evidence):** As noted in our response to Reviewer Lykw (Q1), scaling up the data (Stage II & III) provided only a **marginal gain of +1.1%** (51.2% to 52.3%). In contrast, our **core methods outperformed the BCE baseline by 34.5%** under the same settings (16.7% to 51.2%). This proves that our results are driven by methodological innovation in fair settings rather than a reliance on data scaling. Besides, data scaling and RL are general, necessary operations for training standard large MLLMs, rather than task-specific tricks of ours.
> > >
> > > | Method | BCE | Indiv. Mean | Rel. Sample | Data Scale | RL | Fine-tune | Results |
> > > | :--- | :---: | :---: | :---: | :---: | :---: | :---: | :---: |
> > > | BCE (our baseline, from stage I) | ✓ | | | | | | 16.7 |
> > > | + Indiv. Mean (ours, from stage I) | ✓ | ✓ | | | | | 32.7 |
> > > | + Rel. Sampling (ours, from stage I) | ✓ | ✓ | ✓ | | | | 51.2 |
> > > | + Data Scale (after Stage II & III) | ✓ | ✓ | ✓ | ✓ | | | 52.3 |
> > > | + RL (stage IV) | ✓ | ✓ | ✓ | ✓ | ✓ | | 54.2 |
> > > | + Extra Fine-tune (optional) | ✓ | ✓ | ✓ | ✓ | ✓ | ✓ | 55.2 |
> > >
> > > ---
> > >
> > > **2. Clarification on Title and Descriptions**
> > >
> > > Regarding the consistency of our title, as stated in our TL;DR: "we enable standard MLLMs to directly perform dense prediction tasks via multi-label vision supervision, eliminating the need for extra task-specific decoders." We do not claim that existing MLLMs are already dense predictors. Instead, we emphasize that with appropriate supervision, a standard architecture can achieve dense predictions by leveraging its intrinsic vision tokens. To accurately reflect this and avoid misunderstanding, we have planned the following updates:
> > >
> > > *   **Refined Title**: In order to soften the phrasing as suggested, we will update the title to: **"DenseMLLM: Standard Multimodal LLMs for Dense Prediction"**. This change makes the title more concise while avoiding potential misunderstandings regarding the model's inherent properties.
> > > *   **Clarifying "Intrinsic" Capabilities**: In the Abstract and Introduction, we will refine our statements to emphasize: **"standard MLLM architectures utilizing intrinsic vision tokens for dense predictions without extrinsic decoders."** This clarifies that our method eliminates the need for task-specific extrinsic modules by leveraging the tokens already present in the architecture.
> > > *   **Preserving the "Standard" Architecture**: We will highlight that DenseMLLM maintains a standard architecture. Our model will be **compatible with the standard HuggingFace `transformers` library**, meaning it can be loaded and run in a standard way without any custom operators or architectural modifications.
> > >
> > > ---
> > >
> > > Since the concerns regarding title consistency and the core capability are being addressed, we would be grateful if you would consider increasing the score in light of these updates. Thank you again for your time.

---

### Official Review · Reviewer_PTSn · 2026-03-12

**Soundness:** 3
**Presentation:** 4
**Significance:** 4
**Originality:** 4
**Overall Recommendation:** 6
**Confidence:** 4

**Summary:**

The authors tackle tasks like image segmentation, depth detection, and Referring Expression Segmentation, tasks that involve per-pixel classification. They introduce a method that uses the standard architecture of MLLMs but utilizes the vision token logits to perform these tasks. This way, there is no need for a task-specific external decoder or classification head, which means the model can be used without modification on any task by simply changing the instructions. The model (specifically the language part) first determines the categories (tokens) related to the user instructions. For example, when the prompt asks to segment a cat and a dog, the model determines `<ref>dog</ref><ref>cat</ref><ref>BG</ref>` as the target categories. Then, for each of the vision tokens, it applies an argmax operation over the given categories to determine the correct category.

For training, they use a multi-token objective since each pixel (patch) does not belong to only one category. For example, from a depth perspective, a pixel belongs to a certain category, and it might simultaneously belong to another semantic category. Therefore, their objective can work with multiple target categories. To incorporate this, they remove the softmax function and treat each logit as an independent binary probability. In this new formalization, there are many negative labels for each logit and only a few positive labels, making the training unbalanced. To solve this issue, they use a top-k approach and focus only on the top-k negative tokens in addition to positive tokens.

For training, they first pre-train their MLLM using 10T language tokens and 1.8T multimodal alignment tokens. In the next stage, they use the training technique explained above. In the final two training stages, they apply SFT and DAPO.

**Compliance With Llm Reviewing Policy:**

Affirmed.

**Final Justification:**

The autors fully addressed all my concerns. Their paper is very interesting for the community in my opinion. The scale of traning and experiments are great. The trained model works very well and is of its own interest. Hence I give strong accept.

**Key Questions For Authors:**

1. The compute required for training is not specified. Could you provide these details?
2. I am very interested in the codebook phase at the end of Stage 1. Could you provide more details regarding its implementation and objectives? (also see Weakness 2)

**Limitations:**

Yes

**Strengths And Weaknesses:**

## Strengths

1. Training a SoTA model.
2. Interesting training method.
3. The model can be used for any image segmentation task out of the box.
4. The paper is written clearly.

## Weaknesses

1. Training data is discussed minimally.
2. The codebook part at the end of Stage 1 is interesting but lacks detail. What is the goal of that stage of training, and how is it executed? Does the codebook use the same token space as the language model, or are the token IDs separate? Is the goal for each token to predict its own codebook ID, or the next image patch's codebook ID? These details are entirely missing.
3. Stages 3 and 4 both lack sufficient detail.

---

> ### Author Rebuttal · Authors · 2026-03-28
>
> We sincerely thank the reviewer for recognizing the value of our work and for providing such an accurate, professional, and constructive review. We completely agree that comprehensive details regarding the training data and procedures are crucial for transparency and reproducibility.
>
> Regarding your requests for more specifics on the training data and the detailed implementations (particularly the codebook phase), we will release a link to our open-source project upon acceptance. Within this repository, we will provide a comprehensive 76-page **technical report for more details** about training pipelines, data configurations, and compute details. Most importantly, it contains an extensive and in-depth elaboration on the complete implementation, objectives, and hyper-parameters of the codebook mechanism. Below is the detailed point-by-point response:
>
> ---
>
> ### Q1: Training Data Discussion
>
> **Concern:** Provide more details about training data and compute required for training.
>
> **Response:**
> *   **Compute:**  Thank you for your suggestion. As noted in Line 703, our ablation studies and ADE20k fine-tuning are highly efficient, requiring only 186 GPU hours (8 GPUs for 5.6K iterations), which facilitates easy validation for specific tasks. For the foundational pre-training (Stages I & II), the model was trained on 11.8T and 0.6T tokens respectively. While specific hardware cluster details are subject to institutional confidentiality policies, the total compute budget is on par with training a standard foundational model of similar scale (e.g., Qwen3VL-4B). For the downstream Stage III, the training cost scales linearly with the number of tasks; for instance, training on five datasets of similar size to ADE20k requires approximately 5x186 GPU hours.
> *   **Data:** We will release the project with a technical report that provides more details on each involved tasks. The data details involves data sources, data curation, and data pipelines related to dense prediction and other tasks. This report spans roughly eight pages with comprehensive data details, which is comprehensive and detailed but too long to write in this rebuttal.
>
> ---
>
> ### Q2: Details on the Codebook
>
> **Concern:** Details on the codebook in Stage1 including the training goal and token type.
>
> **Response:**
> We thank the reviewer for their keen interest in the Stage 1 Codebook phase. We address your specific questions below. For more comprehensive details, we have dedicated approximately three pages to discussing this component in the report.
> *   **Training Goal and Execution:** Pre-training supervision on vision tokens aims to provide dense semantic visual supervision. This enables the model to treat visual signals as supervisory targets rather than passive conditions, ensuring the preservation of fine-grained information typically discarded by standard text-generation objectives.
> *   **Token Space Setting:** Token IDs are incorporated into the text vocabulary as discrete vision tokens, establishing a unified vocabulary set. Central to this design is a synergistic vision tokenizer that fuses high-level semantic concepts (features from  SigLIP-2) with low-level geometric structures (from DINOv3) to produce discrete codes using the index backpropagation quantization, thereby facilitating dense semantic visual supervision. The added codebook size is 150,000.
> *   **Prediction Target:** We predict the codebook ID of the subsequent image patch via standard the next-token prediction paradigm and loss. This paradigm ensures seamless compatibility with standard MLLMs and facilitates simultaneous supervision across both visual and textual modalities.
>
> ---
>
> ### Q3: Further Details on Training Stages
>
> **Concern:** Providing further details regarding Stages 3 and 4.
>
> **Response:**
> Thank you for your suggestion. Apart from Appendix B, we will include more details in the open-source link regarding the technical aspects, particularly the training processes and data-related details of Stage 3 and Stage 4. For example, we introduced how we train for general VQA tasks in Stage 3 and give reward details in RL for tasks out of dense prediction. The link will be provided in the paper and on OpenReview after acceptance, in compliance with the double-blind review guidelines.

---

> > ### Author Rebuttal · Reviewer_PTSn · 2026-03-31
> >
> > If my concerns are fully addressed. The fact that they are going to publish all the details, including the training dataset they used, is a very positive factor. I will hence increase my score to strong accept. I think the paper is very interesting for the vision community and could be highly impactful.

---

> > > ### Author Response · Authors · 2026-04-01
> > >
> > > We are truly grateful to the reviewer for the final feedback and for the decision to raise the score. We are pleased that our clarifications on the training data, the codebook mechanism, and the subsequent training stages have fully addressed your initial concerns.
> > >
> > > We share your view that transparency is vital for the community. We would like to reaffirm our commitment to making the open-source project available with a technical report about more details upon acceptance. This will include all the granular details regarding data pipelines and implementation parameters that you highlighted. We thank you for your insightful suggestions which helped us better emphasize the importance of these details.

---

### Official Review · Reviewer_Lykw · 2026-03-13

**Soundness:** 3
**Presentation:** 2
**Significance:** 3
**Originality:** 3
**Overall Recommendation:** 5
**Confidence:** 4

**Summary:**

This paper argues that, with appropriate supervision, the vision tokens intrinsically produced by a standard MLLM already constitute a powerful, architecture-free intrinsic dense predictor. Based on this, this paper introduce DenseMLLM, a general-purpose MLLM that accommodates dense prediction tasks without relying on task-specific decoders. DenseMLLM includes two core designs: 1) Standard MLLM for dense prediction and 2) Vision NTP for multi-label. Experimental results show that it outperforms VisionLLM-v2, DepthLM and UFO on semantic segmentation, depth estimation and anaphora segmentation respectively. Meanwhile, it matches Qwen3-VL-4Bon standard general VL benchmarks. This indicates that DenseMLLM not only possesses the capabilities of a general visual-language model but also has a certain competitive ability in dense visual understanding.

**Compliance With Llm Reviewing Policy:**

Affirmed.

**Final Justification:**

The authors basically solved my concern, I will raise the score to 5.

**Key Questions For Authors:**

see weaknesses.
Furthermore, there are few questions:
The description of semantic segmentation in the related work is confusing: “Semantic segmentation assigns vocabularies to every pixel, a task evolved from CNNs (Zhao et al., 2017) to Transformer (Cheng et al., 2022; Xie et al., 2021) and Diffusion architectures (Zhao et al., 2023).” Why did the author choose to start from the selection of model architecture to summarize the semantic segmentation task? What is the connection between this and the method proposed in this paper?
There are also some issues of incorrect citations in the paper. The author should carefully review them to meet the basic academic standards, such as:
In the main body section："...polygon-based methods like VisionLLM (Lu et al., 2024) often lack precision, and textbased approaches like..."
However, the information about VisionLLM in the citation is: "Wang, W., Chen, Z., Chen, X., Wu, J., Zhu, X., Zeng, G., Luo, P., Lu, T., Zhou, J., Qiao, Y., and Dai, J. Visionllm: Large language model is also an open-ended decoder for vision-centric tasks. ArXiv, abs/2305.11175, 2023b. URL https://api.semanticscholar. org/CorpusID:258762579."

**Limitations:**

yes

**Strengths And Weaknesses:**

Strengths
The method proposed in the paper is simple and effective. It directly reads the dense prediction results from the vision tokens instead of attaching additional segmentation/depth heads.
The experiment had a wide coverage. This paper not only evaluated semantic segmentation, depth estimation, and object representation segmentation, but also assessed general visual language abilities such as universal VQA, OCR, and reasoning. This well supported the author's core argument that "dense capabilities do not undermine general capabilities".

Weaknesses
The performance of the paper largely depends on a complex four-stage training process, large-scale data mixing, SFT and RL. Therefore, it is not yet clear whether it is the core methods proposed in the paper itself or the large-scale training recipe that truly leads to a significant improvement in capabilities.
The paper does not clearly explain the specific datasets used in the training stages, especially how the training and test datasets are isolated to ensure there is no problem of test data leakage?
The fairness of the paper's comparison with baseline models remains questionable. Many of the comparison methods in Table 1 have significant differences in model size, visual encoder, training data, and whether they are single-task fine-tuned. Therefore, although the advantage of "no additional modules" is obvious, the performance comparison may not be completely fair, especially when comparing with some stronger specialist models. In particular, compared with some stronger specialist models, DenseMLLM has not yet fully dominated.
Although DenseMLLM has certain dense visual understanding capabilities, appropriately supplementing the visualization results of the model's failure cases in dense visual tasks will be beneficial for readers to understand the limitations of the method.
The paper's claim is that "Standard Multimodal LLMs are Intrinsic Dense Predictors", but from the experiments, this ability seems more like being "shaped" through specialized supervision, large-scale data and training strategies, rather than being inherently possessed by standard MLLM. Is the current title and discussion perhaps too strong?

---

> ### Author Rebuttal · Authors · 2026-03-28
>
> We sincerely thank the reviewers for the constructive feedback. To state our conclusions directly: Our massive performance leaps stem strictly from our core algorithmic designs, and comprehensive ablations fully guarantee both the effectiveness of our method and the fairness of our baseline comparisons. Furthermore, we will release the project with more data details, alongside thoroughly refined writing and citations. Below is the detailed point-by-point response:
>
> ---
>
> ### Q1: Impact of Core Methodology vs. Training Recipe
> **Concern:** Does the improvement stem from the core methodology or the large-scale training recipe?
>
> **Response:**
> **Our core methodology is the absolute primary driver of the performance.** As demonstrated in Table 3 (below), starting from Stage I, our two core designs (Indiv. Mean + Rel. Sampling) yield a massive **+34.5%** absolute improvement (from 16.7% to 51.2%).
>
> In contrast, simply scaling up the data (Stage II & III) and applying RL (Stage IV) only provide marginal gains (+1.1% and +1.9%, respectively). **It is important to note that data scaling and RL are general, necessary operations for training standard large MLLMs, rather than task-specific tricks of ours.** The core dense prediction capability is fundamentally unlocked by our method.
>
> | Method | BCE | Indiv. Mean | Rel. Sample | Data Scale | RL | Fine-tune | Results |
> | :--- | :---: | :---: | :---: | :---: | :---: | :---: | :---: |
> | BCE (our baseline, from stage I) | ✓ | | | | | | 16.7 |
> | + Indiv. Mean (ours, from stage I) | ✓ | ✓ | | | | | 32.7 |
> | + Rel. Sampling (ours, from stage I) | ✓ | ✓ | ✓ | | | | 51.2 |
> | + Data Scale (after Stage II & III) | ✓ | ✓ | ✓ | ✓ | | | 52.3 |
> | + RL (stage IV) | ✓ | ✓ | ✓ | ✓ | ✓ | | 54.2 |
> | + Extra Fine-tune (optional) | ✓ | ✓ | ✓ | ✓ | ✓ | ✓ | 55.2 |
>
> ---
>
> ### Q2: Dataset Details and Data Isolation Protocols
> **Concern:** Clarification is needed regarding the dataset details and the isolation protocols.
>
> **Response:**
> We employ strict data isolation. After image deduplication via md5sum, we ensure that no test set images or annotations appear in the training data. In the next version, we will provide a link to release this project with a 76-page technical report, which exhaustively details the data composition, processing pipelines, and our rigorous data-separation protocols.
>
> ---
>
> ### Q3: Clarifications on Baseline Comparisons in Table 1
> **Concern:** Differences in setups complicate baseline comparisons, especially against specialist models.
>
> **Response:**
>
> - ***Fairness***: Table 1 includes five different settings, with the best results in each setting highlighted in bold. Under the standard MLLM setting, our results are the best, with model parameters comparable to or fewer than other MLLMs (4B vs. 3B/4B/7B), demonstrating clear fairness. Consistency across different settings cannot be guaranteed, for example, specialist and CLIP models, being earlier-stage, do not utilize large 4B models, making direct parameter-level comparison infeasible.
>
> - ***Specialist Models***: Most specialist models are single-task fine-tuned (gray in Table 1) and heavily benefit from specialized training and architecture (e.g, FPN, multi-scale, decoders). Even though our general-purpose model without task-specific fine-tuning still outperforms them on COCOstuff, VOC20, and RefCOCO, etc, matches them on others. Besides, our method excels across other four settings, firmly validating its superiority.
>
> ---
>
> ### Q4: Visualizations of Challenging Cases
> **Concern:** It would be beneficial to supplement visualizations with challenging/failure cases.
>
> **Response:**
> Thank you for the valuable suggestion. We will include a dedicated section of challenging and failure cases in the Appendix to transparently discuss the model’s boundaries and encourage further improvements.
>
> ---
>
> ### Q5: Discussion on the Concept of "Intrinsic Dense Predictors"
> **Concern:** The title/claims might be too strong; the ability is shaped by training rather than inherent.
>
> **Response:**
> The title means “DenseMLLM enables standard multimodal LLMs to be intrinsic dense predictors” (see the TL;DR). While your understanding that our training strategies enable this is correct, we do not claim that existing standard models are already dense predictors, highlighting the necessity and value of our approach.
>
> ---
>
> ### Q6: Writing about Related Work and Citation
> **Concern:** The connection between the evolution of segmentation architectures and the proposed method; Formatting and mismatch issues in citations.
>
> **Response:**
> We introduced the architectural development to highlight that using MLLMs for dense prediction differs from previous architectures (e.g., CNNs/Transformers) and remains a pioneering effort. We will refine it to make the connection clearer. We also thank the reviewers for their careful reading. In the next version, we will thoroughly review and correct all citation mismatches and formatting issues in the references.

---

> > ### Author Rebuttal · Reviewer_Lykw · 2026-04-03
> >
> > The rebuttal usefully points to Table 3 and argues that the major gain comes from the proposed methodological components (Indiv. Mean and Rel. Sampling), while larger-scale data and RL contribute only relatively small additional improvements. I appreciate the authors’ willingness to add challenging and failure cases in the appendix. This would improve the paper and help readers better understand the model’s limitations.
> > Although the rebuttal clarifies the intended interpretation of the title, I still find the current wording somewhat stronger than what is fully supported by the evidence. The demonstrated capability appears to be enabled by the proposed supervision and training design, rather than being an inherent property of standard MLLMs without such adaptation. I would still recommend softening or sharpening this phrasing in the final version.
> > Overall, the rebuttal improves my confidence in the technical contribution, especially regarding the importance of the proposed vision-token supervision and sampling strategy.

---

> > > ### Author Response · Authors · 2026-04-04
> > >
> > > We sincerely thank you for your constructive feedback and are pleased that our response has fully resolved your concerns. We especially appreciate your insights regarding the title and the nuance of our claims. We will incorporate your suggestions into the final version to ensure the paper's positioning is both accurate and well-supported. Below is our plan for the revision:
> > >
> > > *   **Refined Title**: In order to **soften the phrasing** as you suggested, we will update the title to: **"DenseMLLM: Standard Multimodal LLMs for Dense Prediction"**. This change makes the title more concise while avoiding potential misunderstandings regarding the model's inherent properties.
> > >
> > > *   **Clarifying "Intrinsic" Capabilities**: In the Abstract and Introduction, we will refine our statements to emphasize: **"standard MLLM architectures utilizing intrinsic vision tokens for dense predictions without extrinsic decoders."** By specifying that "intrinsic" refers to the vision tokens, we clarify that our method eliminates the need for task-specific extrinsic modules.
> > >
> > > *   **Preserving the "Standard" Architecture**: We will highlight that DenseMLLM maintains a standard architecture. Our model will be **compatible with the standard HuggingFace `transformers` library**, meaning it can be loaded and run in a standard way without any custom operators or architectural modifications.
> > >
> > > Since you mentioned that your concerns are fully resolved, we would be grateful if you would consider increasing the score. Thank you again for your time and for your valuable help in improving this paper.

---

### Decision · Program_Chairs · 2026-04-30

**Decision:**

Accept (regular)

**Comment:**

This submission introducing DenseMLLM, a novel approach that enables standard multi-modal large language models (MLLMs) to perform dense prediction tasks using intrinsic vision tokens, eliminating the need for task-specific decoders. The proposed method leverages the standard MLLM architecture, utilizing vision token logits to execute tasks, thus allowing the model to be adapted to any task by merely modifying the instructions. This approach is characterized as simple, effective, and versatile, with demonstrated applicability across dense perception and general visual-language tasks, successfully integrating dense capabilities without compromising general performance.

The submission received a positive average score, accompanied by a strong accept rating from Reviewer PTSn, who highlighted its potential impact on the vision community. The experiments design and results are sound and convincing. The authors were engaged in the rebuttal period have successfully addressed the concerns from other reviewers, we recommend to accept this submission.